Brief Communication

# One-carbon fixation via the synthetic reductive glycine pathway exceeds yield of the Calvin cycle

Beau Dronsella ●[1,2] ✉, Enrico Orsi ●[1], Helena Schulz-Mirbach[2], Sara Benito-Vaquerizo[3,4], Suzan Yilmaz[5], Timo Glatter ●[6], Arren Bar-Even ●[1], Tobias J. Erb ●[2,7] & Nico J. Claassens ●[5] ✉

One-carbon feedstocks such as formate could be promising renewable substrates for sustainable microbial production of food, fuels and chemicals. Here we replace the native energy-inefficient Calvin–Benson–Bassham cycle in *Cupriavidus necator* with the more energy-efficient reductive glycine pathway for growth on formate and $CO_2$. In chemostats, our engineered strain reached a 17% higher biomass yield than the wild type and a yield higher than any natural formatotroph using the Calvin cycle. This shows the potential of synthetic metabolism to realize sustainable, bio-based production.

A promising strategy to realize higher microbial yields on renewable, electrochemically derived formate is the implementation of more ATP-efficient, synthetic pathways[1]. One of the most promising synthetic pathways with potential to show this yield improvement is the reductive glycine pathway (rGlyP), which is the most ATP-efficient aerobic pathway for formate assimilation (Fig. 1a,b). We recently demonstrated full synthetic pathway operation, partly expressed from plasmids, via modular and evolutionary engineering in *Cupriavidus necator*, but showed that biomass yields were so far still not exceeding the upper bounds of natural formatotrophy via the Calvin–Benson–Bassham (CBB) cycle[2]. Here we show the full integration of the synthetic rGlyP into the genome of *C. necator*. In chemostat experiments, we demonstrate a yield of 4.52 g cell dry weight (CDW) mol$^{-1}$ formate for the engineered strain. This yield is 17% higher than that of the wild type and higher than any reported yield for growth on formate via engineered pathways as well as the CBB cycle. This study shows that superior microbial growth yields via synthetic one-carbon (C1)-assimilation pathways are feasible, paving the way for more efficient, sustainable bioproduction.

To generate a genome-integrated rGlyP strain, we first cured the previously engineered *Cupriavidus* reductive glycine 4 (CRG4) strain[2] of the plasmid expressing the first (C1) module of the rGlyP (Extended Data Fig. 1 and Methods). As this strain still overexpressed the remaining C2 and C3 modules of the rGlyP, it was an ideal platform to select for optimal, genomically expressed C1 module activity on formate. We then integrated a library of operons expressing the C1 module from different strength promoters into the genome using the Tn5-transposon machinery and subsequent selection for fast growth of clones on formate minimal media (Extended Data Fig. 1 and Supplementary Note 1). A fast-growing strain was selected (CRG5).

Next, we also integrated the C3 module that was still expressed from a plasmid into the genome of CRG5, using the same approach as above (Supplementary Note 2). This led to the selection of a fast-growing CRG6 strain with a completely genome-integrated rGlyP. The strains were characterized for their growth and genome sequences (Extended Data Figs. 2–5 and Supplementary Tables 1–3). Strain CRG6 clearly grew faster than CRG4 at a doubling time in batch cultures of ~11 h, which was (still) slower than that of the wild-type CBB cycle strain (~6 h doubling time; Fig. 2a). However, both CRG5 and CRG6 consistently grew to a higher maximum biomass optical density than both CRG4 and the wild type (Fig. 2a and Extended Data Fig. 2).

To investigate and compare the cellular proteome allocation in the rGlyP strains CRG4 (plasmid expressed) and CRG6 (genomically

[1]Systems and Synthetic Metabolism, Max Planck Institute of Molecular Plant Physiology, Potsdam, Germany. [2]Biochemistry and Synthetic Metabolism, Max Planck Institute for Terrestrial Microbiology, Marburg, Germany. [3]Laboratory of Systems and Synthetic Biology, Wageningen University, Wageningen, The Netherlands. [4]Genome Biology Unit, European Molecular Biology Laboratory, Heidelberg, Germany. [5]Laboratory of Microbiology, Wageningen University, Wageningen, The Netherlands. [6]Core Facility for Mass Spectrometry and Proteomics, Max Planck Institute for Terrestrial Microbiology, Marburg, Germany. [7]Center for Synthetic Microbiology, Marburg, Germany. ✉e-mail: beau.dronsella@mpi-marburg.mpg.de; nico.claassens@wur.nl

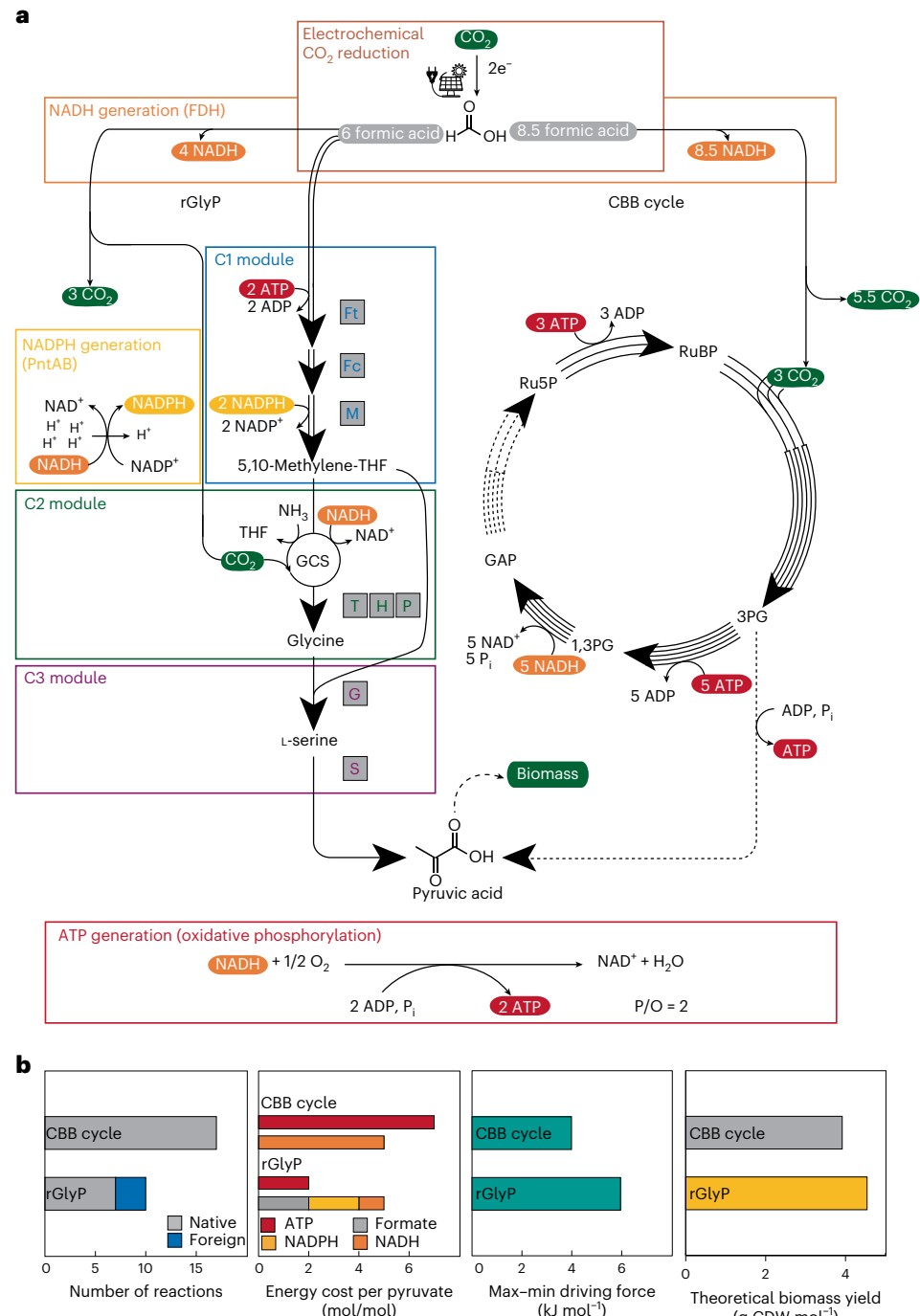

**Fig. 1 | The rGlyP compared with the CBB cycle. a**, In the $CO_2$ module, formate is generated from electrochemical reduction of $CO_2$. Formate can then be used to form pyruvate as indicated by the rGlyP or the CBB cycle, resulting in different formate requirements. In the rGlyP, the C1 module activates and reduces formate to 5,10-methylene-THF via three heterologous enzymes from *M. extorquens*: FtfL (Ft), FchA (Fc) and MtdA (M). Next, methylene-THF is converted to glycine by the glycine cleavage system (GCS, composed of the enzymes GcvT, H and P) operating in the reductive direction (C2 module). Finally, glycine is condensed with another 5,10-methylene-THF via serine hydroxymethyltransferase (GlyA, G) to yield serine, which is then dehydrated by serine deaminase (SdaA, S) to pyruvate. NADH is regenerated from formate via formate dehydrogenase. NADPH is regenerated by proton-translocating, membrane-bound transhydrogenase (PntAB). Formate and pyruvate are referred to as such in the text, but are here depicted in their protonated forms, in which they are metabolized. In the CBB cycle, three ribulose-5-phosphate molecules and $CO_2$ are converted into 3-phosphoglycerate via the CBB cycle signature enzymes phosphoribulokinase (Prk) and RuBisCO. Subsequently, one of six generated 3-phosphoglycerate molecules can be used in metabolism, for example, via conversion to pyruvate. The other five 3-phosphoglycerate molecules are recycled into ribulose-5-phosphate via gluconeogenesis and the pentose phosphate pathway. For formatotrophic growth via the CBB cycle, all formate is oxidized into NADH and $CO_2$ to supply energy and carbon to the CBB cycle. **b**, Comparison of the rGlyP with the CBB cycle for the number of enzymatic reactions required, cost of ATP and reducing equivalents, minimal thermodynamic driving force MDF (Supplementary Fig. 1) and biomass yield predicted by the model at a doubling time of 14 h and assuming GAM = 135 mmol gCDW$^{-1}$ and NGAM = 3 mmol gCDW$^{-1}$ h$^{-1}$. Values were calculated for both routes from formate to pyruvate at 10% $CO_2$ (3.4 mM). The GAPDH reaction was set as NAD$^+$-dependent for the CBB cycle, and the standard NADH/NAD$^+$ ratio of 0.1 was used, as the ratio has not been determined for *C. necator* grown on formate.

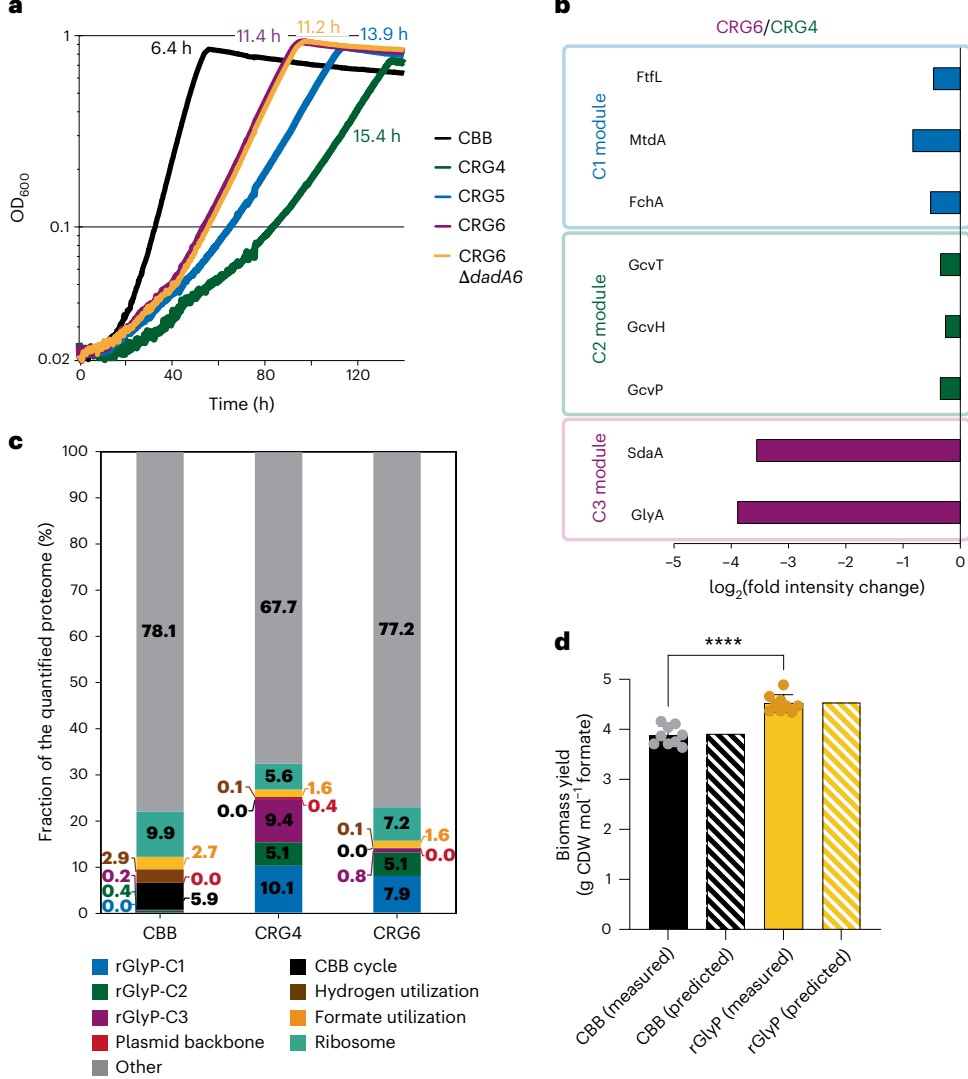

**Fig. 2 | Characterization of the rGlyP and CBB cycle *C. necator* strains.**
**a**, Growth of *C. necator* H16 Δ*phaC1* strains harbouring the CBB cycle compared with that of CRG4, CRG5 and CRG6 in M9 minimal media supplemented with 80 mM formate and 100 mM bicarbonate with 10% CO₂ in the headspace. The doubling time in hours of the strains is presented in the designated strain colour. Curves depict the mean of at least two technical replicates and are representative of three experiments conducted in the same conditions to ensure reproducibility. **b**, Relative protein intensity changes of the rGlyP modules in strain CRG6 relative to CRG4 both grown on formate minimal media. **c**, Fractions of the quantified proteome associated with various metabolic tasks in the strains CBB, CRG4 and CRG6 all grown on formate minimal media. Clustering criteria for grouping of proteins by metabolic tasks are provided in Methods. **d**, Measured biomass

yields in grams CDW per mole formate consumed are shown, both for the CBB cycle (*C. necator* Δ*phaC1*) and rGlyP (CRG6 Δ*dadA6*) strains grown in formate minimal media in bioreactors in chemostat mode at a dilution rate of 0.05 h⁻¹ (14 h doubling time). Predicted biomass yields are shown in a shaded pattern and are derived from FBA simulations run with a fixed growth rate corresponding to a doubling time of 14 h and maintenance costs of GAM = 135 mmol gCDW⁻¹ and NGAM = 3 mmol gCDW⁻¹ h⁻¹. Measured data represent samples (*n* = 9) obtained during steady state, each having a volume of 50 ml or 100 ml taken over the course of 3 days. All data points and their mean value are given. Error bars indicate standard deviation. Significance was tested via two-tailed unpaired *t*-test. ****$P = 1.33 \times 10^{-6}$; 95% confidence interval = 0.4600–0.8244; difference between means ± s.e.m. = 0.6422 ± 0.08594; d.f. = 8.

expressed), as well as the CBB cycle strain, we performed quantitative proteomics (Fig. 2b,c, Supplementary Note 3 and Extended Data Fig. 6a–c). This analysis showed that the proteome allocation of the C1 module after genome integration decreased only slightly, whereas the allocation of the C3 module was reduced more than tenfold, suggesting that the latter module was redundantly high expressed in CRG4.

High C1 module expression was achieved via transposon insertion downstream of the native *phaP1* promoter, which decreased *phaP1* expression at the same time (Source Data). We confirmed that deletion of *phaP1* alone did not allow for improved growth rate and yields (Extended Data Fig. 7). Overall ~13% of the proteome allocation associated with the rGlyP in CRG4 was freed up in CRG6, which probably allowed allocation of proteome resources to other processes such as

ribosome biosynthesis (Fig. 2c), allowing for the increased growth rate of the CRG6 strain.

We next investigated at two critical rGlyP bifurcation points (formate and glycine) whether wasteful oxidative sinks were limiting biomass formation using in vivo formate dehydrogenase inhibition assays and deletion of glycine oxidase (Δ*dadA6*) (Fig. 2a, Supplementary Note 4 and Extended Data Figs. 2 and 8a–e), but did not observe further improvements in growth rate or yield. This probably reflects that the cells already properly balanced formate assimilation flux with energy generation.

With the seemingly energetically optimal CRG6 Δ*dadA6* strain, we now wanted to accurately determine whether the rGlyP could also support a higher biomass yield on formate than the CBB cycle. Other

than energy consumption differences in the pathway, also growth rate differences between the strains may impact yields. Typically, faster growth rates decrease the relative impact of maintenance energy requirements, which can further increase biomass yield. Furthermore, for formatotrophic growth of *C. necator*, a negative effect on yield has been reported due to high residual formate levels in the medium[3], as formate is generally a toxic substrate that can inhibit growth rates and yields, especially at higher concentrations.

Hence, we decided to accurately determine the formatotrophic yield of the CRG6 Δ*dadA6* and CBB cycle strains by using bioreactors in chemostat mode at the same dilution or growth rate (0.05 h$^{-1}$, that is, a doubling time of ~14 h). The same dilution rate was used in a previous study that characterized the yield via the CBB cycle on formate for *C. necator*, which reported a yield of ~4 g CDW mol$^{-1}$ (ref. 3). The chemostat cultivation also ensures that the formate levels stay low (below the high-performance liquid chromatography (HPLC) detection limit) and hence prevents reductions in yield or growth rate due to toxicity.

In chemostat bioreactor experiments, we obtained biomass yields of 3.88 ± 0.19 g CDW mol$^{-1}$ for the CBB cycle strain and 4.52 ± 0.17 g CDW mol$^{-1}$ for the rGlyP strain, showing a 16.6% higher yield compared with the wild type in the same condition (Fig. 2d). The measured yields are in accordance with the theoretically predicted yield by a genome-scale metabolic model (GEM) of *C. necator* for both the CBB cycle (3.92 g CDW mol$^{-1}$) and the rGlyP (4.54 g CDW mol$^{-1}$) at 14 h of doubling time. This confirms that the energetically superior formatotrophic pathway indeed can support the predicted yield increase.

At the end of the chemostat experiment, the dilution rate was increased to 0.087 h$^{-1}$, which corresponds to 8 h of doubling time. Both strains were maintained at this dilution rate (Extended Data Fig. 9), which for the rGlyP strain marks the fastest measured growth of *C. necator* strains growing via the rGlyP so far. At a dilution rate of 0.1 h$^{-1}$, however, the CRG6 Δ*dadA6* strain was washed out, indicating that it cannot yet sustain the same growth rate as the CBB-employing wild type (fastest doubling time ~4 h)[3]. The lower growth rate of the rGlyP strain compared with the CBB strain could still limit bioproduction rates, but probably can be further improved by long-term evolution as shown recently for the wild-type strain[4].

Our study shows that an engineered C1-assimilation pathway can outcompete the biomass yield of a natural pathway in the same organism. In fact, the measured formatotrophic yield of 4.52 g CDW mol$^{-1}$ is, to our best knowledge, higher than any so far reported yield on formate for natural organisms using the CBB cycle, as well as for engineered formatotrophs (Supplementary Table 4). Notably, the relative yield increase of ~17% can in certain conditions even be higher, as in our comparison experiments the CBB cycle is not performing wasteful oxygenation and photorespiration (due to the high $CO_2$).

The measured yield, being very similar to the theoretically predicted yield, indicates that the metabolic network of the engineered strain including the synthetic assimilation pathway is well performing. This suggests that there is no major remaining losses or energy-wasting metabolic processes. This is in contrast to the so far demonstrated low yields for several recently engineered synthetic *Escherichia coli* and *Saccharomyces cerevisiae* formatotrophs and methylotroph*s* (Supplementary Table 4). This is probably because the metabolic networks of the naturally strict heterotrophs are less suited for autotrophic metabolism. An example is the relatively high remaining activity of the tricarboxylic acid (TCA) cycle in the *E. coli* strain growing via the rGlyP[5]. During formatotrophic growth, the TCA cycle is a redundant process, wasting acetyl-CoA and hence decreasing yields on formate. *C. necator* seemingly already evolved to suppress TCA cycle activity efficiently during formatotrophic and autotrophic growth with the CBB cycle[6], and this regulation seems sustained in the rGlyP strains as indicated via $^{13}C$-labelling in our previous study[2].

Our study shows that designed pathways with theoretically higher efficiencies can indeed improve yields in vivo. In the future, other promising energy-efficient, synthetic C1-fixation pathways, which were only shown in vitro such as the synthetic crotonyl-CoA-ethylmalonyl-CoA-hydroxybutyryl-CoA (CETCH) cycle for $CO_2$ fixation[7], may outperform the CBB cycle in vivo, once successfully established. Implementation of such $CO_2$ fixation pathways in autotrophic hosts, such as *C. necator* or photosynthetic microorganisms, may similarly benefit from the natively already well-wired autotrophic metabolic network to efficiently realize higher yields than the CBB cycle, as observed here. The (randomized) transposon integration technique and growth-coupled selection approach in this study could furthermore serve as a blueprint to realize efficient pathway implementation, also in more difficult-to-engineer bacteria, as well as eukaryotes, such as yeast and microalgae. Realizing higher yields and faster growth on formate, $CO_2$ and other C1 substrates will aid to make the electro-microbial production route an attractive, economically feasible reality for sustainable production of carbon-based chemicals, fuels, feed and food.

## Methods

### Bacterial strains and conjugation

A complete list of strains and plasmids used in this study can be found in Supplementary Tables 5 and 6. *C. necator* H16 deleted in polyhydroxybutyrate biosynthesis (Δ*phaC1*) served as the base strain ('wild type') in this work. All CRG strains are also deleted for carbon fixation via the CBB cycle by deletion of both ribulose-1,5-bisphosphate carboxylase/oxygenase (RuBisCO) subunits on chromosome 2 and megaplasmid (Δ*cbbSL2*, Δ*cbbSLp*). Routine cloning was performed in *E. coli* DH5α, while *E. coli* ST18 cells were used for conjugation of mobilizable plasmids to *C. necator* via biparental spot mating.

### Cultivation conditions

*C. necator* and *E. coli* were grown on lysogeny broth (LB; 10 g l$^{-1}$ NaCl, 5 g l$^{-1}$ yeast extract and 10 g l$^{-1}$ tryptone) for routine cultivation and genetic modifications. When appropriate, the antibiotics kanamycin (100 µg ml$^{-1}$ for *C. necator* and 50 µg ml$^{-1}$ for *E. coli*), chloramphenicol (30 µg ml$^{-1}$), tetracycline (10 µg ml$^{-1}$), ampicillin (100 µg ml$^{-1}$ for *E. coli*) and gentamycin (20 µg ml$^{-1}$ for *C. necator*) were added. Growth experiments were conducted in M9 minimal medium (47.8 mM $Na_2HPO_4$, 22 mM $KH_2PO_4$, 8.6 mM NaCl, 18.7 mM $NH_4Cl$, 2 mM $MgSO_4$ and 100 µM $CaCl_2$), supplemented with trace elements (134 µM EDTA, 31 µM $FeCl_3 \cdot 6H_2O$, 6.2 µM $ZnCl_2$, 0.76 µM $CuCl_2 \cdot 2H_2O$, 0.42 µM $CoCl_2 \cdot 2H_2O$, 1.62 µM $H_3BO_3$, 0.081 µM $MnCl_2 \cdot 4H_2O$). Routine cultivation was performed in 4 ml medium in 12-ml glass tubes in an orbital shaker incubator at 240 rpm at 30 °C and 37 °C for *C. necator* and *E. coli*, respectively. M9 minimal medium was supplemented with 80 mM sodium formate and 100 mM sodium bicarbonate, with the pH adjusted to 7.2, under a headspace of 10% $CO_2$ (v/v) for formatotrophic growth. Strictly seen enough $CO_2$ is generated intracellularly to drive the GCV carboxylation of the rGlyP by formate oxidation. However, in relatively low-biomass-density cultures, with high aeration, as performed in this study, this may be insufficient. Hence, we supplement sodium bicarbonate in the medium (only during batch cultivations) and $CO_2$ in the headspace or gas supply during the bioreactor cultivation.

No antibiotics were added during growth characterization experiments in the plate reader. Growth measurements were obtained from 96-well-plate experiments (Nunc transparent flat bottom, Thermo Scientific). Strains were typically pre-cultured in M9 minimal medium supplemented with 20 mM pyruvate. Cells were collected, washed twice and inoculated at an optical density at 600 nm (OD$_{600}$) of 0.01. To avoid evaporation while maintaining diffusion of $O_2$ and $CO_2$, 150 µl of cell medium mix was topped with 50 µl transparent mineral oil (Sigma-Aldrich). The 96-well plates were incubated at 30 °C with continuous shaking (alternating between 30 s orbital and 30 s linear)

in a Tecan infinite M200Pro plate reader (Tecan). $OD_{600}$ values were measured every 8 min. Growth data were blanked and converted from plate reader $OD_{600}$ to cuvette $OD_{600}$ by multiplication with a factor of 4.35 via a Matlab script. All growth experiments were repeated at least three times, and the growth curves shown are representative curves of these experiments.

## Plasmid curing

Cells containing plasmids to be cured were propagated in LB media without antibiotics. From each grown passage, 50 colonies were streaked out on LB agar plates. These were then replica plated on LB agar plates containing the antibiotic for which the plasmid would provide resistance and on non-antibiotic-containing plates. Once colonies were obtained that did not grow on the antibiotic-containing plates, the respective colonies from the non-selective plate were investigated via PCR targeting the plasmid to confirm the curing of the clone.

## Tn5 vector construction

The vector pBAMD1-4 was provided by P. I. Nikel. The vector backbone was amplified with primers pBAMD-for (GCGCGGCCGCATAAAATCTCT-GAAGATGTG) for the rGlyP-C1 module or pBAMD-for2 (GCGATGCAT-ATAAAATCTCTGAAGATGTG) for the rGlyP-C3 module and pBAMD-rev (GCGGCTAGCGCCTGAGACACAAAGATGTG) without the antibiotic resistance gene in the cargo module (module between the transposon recognition sites ME1 and ME2), and NotI and NheI restriction sites were attached. The operons *mtdA-fch-ftfL* and *sdaA-glyA* were previously cloned under control of the promoters, from weakest to strongest, $P_{14}$/ $P_{PhaCl}$/$P_3$/$P_4$/$P_2$ and $P_{cat}$/$P_{PhaCl}$/$P_3$/$P_4$, respectively. These were then amplified from existing pSEVA221- and pSEVA331-based expression plasmids respectively with primers C1-for (GCGGCTAGCTCTAGGGCGGCG-GATTTGTC) and C1-rev (GCGCGGCCGCTTGGGGACCCCTGGATTCTC) attaching NotI and NheI restriction sites or C1-for and C3-rev (GACAT-GCATTTGGGGACCCCTGGATTCTC) to attach NheI and NsiI sites. Promoter-FCM/SG operons were cloned into pBAMD vectors using restriction ligation.

All genes of the C1 module were previously placed behind a synthetic RBS designed with an RBS Calculator with a translation initiation rate of 30,000 arbitrary (arb.) units (ref. [8]).

## Gene deletions

The plasmid pLO3-dadA6 was used to delete the *dadA6* gene in the CRG6 strain via allelic replacement based on sucrose counter selection with SacB as described previously[2,9]. For the *phaP1* deletion, we used the recently established 'Self-Splicing Intron-Based Riboswitch' (SIBR) system for Cas9-based counter selection[10]. With the use of HR1_phaP1_F (GAGCAAGCCCGTAGGGGGGGGAACTGGGCATCAG-GAC), HR1_phaP1_R (CTGACATCTAGGCGGCTTTGATAACTGCCTGCG), HR2_phaP1_F (TTCAACGCAGGCAGTTATCAAAGCCGCCTAGATGTCAG) and HR2_phaP1_R (CGCCGCCCTAGACAGCTGGGAGGTCGCTGGC-CTCTTTG), 1-kb flanking arms were amplified and cloned into pSIBR004 together with spacer PhaP1-spacer1 (TGCCAACAACGC-CTACGAGT) or PhaP1-spacer2 (TTTCCGAAGCGATTTCATAC). The deletions were confirmed by colony PCR using the primers dadA6-for (TGGAAGGCTACCCCTACTTC) and dadA6-rev (ATAGAAACTCAGCG-GCTGGC) or phaP1-for (ACGTAGCCGATGCCTG) and phaP1-rev (TAG-GTATCGTCGTCGC) and whole-genome sequencing.

## Tn5-mediated knock-in coupled with liquid selective conditions

*E. coli* ST18 strains harbouring the $P_{14}$/$P_{PhaCl}$/$P_3$/$P_4$/$P_2$-C1 and $P_{cat}$/$P_{PhaCl}$/ $P_3$/$P_4$-C3 constructs in pBAMD-Tn5 vectors served as donor strains. *C. necator* strains CRG4.5 and CRG5.5 served as recipient strains for the conjugation. In a 1.5-ml plastic tube, 100 µl of LB-grown dense overnight cultures of *C. necator* and *E. coli* ST18 (supplemented with 50 µg ml$^{-1}$ 5-amniolevulinic acid (ALA)) were mixed. Of this cell mixture,

100 µl was plated on LB + ALA agar plates and dried for 30 min before incubating overnight at 30 °C. The next day, the grown cell lawn was resuspended in LB and a 100-µl inoculum of an $OD_{600} = 1$ mixture of *C. necator* recipient and *E. coli* ST18 donor cells was used to inoculate 4 ml of liquid M9 minimal medium supplemented with 80 mM formate and 100 mM bicarbonate in 10% $CO_2$ (v/v). When cellular growth was observed and the population reached late log to stationary phase, 1 µl of $OD_{600} = 1$ culture was used to reinoculate into 4 ml of selective medium. This population was passaged 10 times to allow the Tn5 transformants with better pathway expression to overtake the population (Extended Data Fig. 3). After passage 10, the population was dilution streaked two consecutive times to single colony on LB agar plates with 20 µg ml$^{-1}$ gentamicin. The isolated single clones were compared with the populations for growth behaviour in the selective formate media and saved in 25% glycerol at −80 °C (Extended Data Fig. 4).

## Bioreactor experiments and biomass yield determination

Bacterial strains were streaked out on LB agar plates containing antibiotics when appropriate. Single clones were inoculated from the plate into 12-cm glass tubes containing 4 ml of M9 minimal medium supplemented with 80 mM formate and 100 mM $HCO_3$ and cultivated at 10% $CO_2$, 30 °C and 250 rpm. Growing cultures were then used to inoculate 50 ml of the same media in 250-ml flasks and cultivated in the same conditions. Pre-cultures were then used to inoculate 1:10 in M9 80 mM formate media and grown in batch at 30 °C and 400 rpm in 400 ml volume in 1 l DASGIP bioreactors (Eppendorf). The reactors were sparged at a constant flow of 60 sl h$^{-1}$ (2.5 volume gas per volume culture per min) with a gas mixture of 10% $CO_2$ and 90% air. Dissolved oxygen was measured online and was maintained above 30% via stirrer speed control (400–800 rpm).

Following batch growth, the OD was increased using fed batch by adding 4 ml or 8 ml of 4 M formic acid once per day. After an $OD_{600}$ of >1.2 was reached, the dilution of the reactor was started with M9 80 mM formate media (pH = 3.7) by setting the pumps to a flow rate of 20 ml h$^{-1}$. The waste pump flow rate was set to 100 ml h$^{-1}$. The waste pipe was positioned at the surface level of the 400 ml culture volume. The lag phase of the CRG6 Δ*dadA6* strain caused the pH to quickly drop, as formate could not be oxidized fast enough, which resulted in a positive feedback loop of lower pH and more lag. Hence, pH-neutral M9 80 mM formate was supplied at the same dilution rate of 0.05 h$^{-1}$ with 1 M formic acid being used to maintain the pH at 7.1. After the strain reached steady state, the medium was switched to the low-pH M9 80 mM formate medium, which during exponential growth did not cause a wash-out of the strain. Steady state for this medium was reached after more than 7 reactor turnovers (140 h). Every day for 3 days, 50- and 100-ml samples were taken. Formate-grown cells were collected by centrifugation at 3,220 g for 20 min and washed 3 times in 50 ml double distilled water to remove residual medium components. Washed cells were then pipetted into custom-made pre-dried and pre-weighed aluminium trays. The samples were dried for a period of 24 h at 90 °C and were then weighed to obtain the additional weight from the dry cells.

Formate concentrations were quantified through HPLC using the Shimadzu LC-2030C Plus system equipped with a Shodex SH1821 column (8 mm × 300 mm, at 45 °C), using refractive index detection. Elution was performed with 0.01 N sulfuric acid at a flow rate of 1 ml min$^{-1}$, and 10 µl of internal standard (0.1 M sulfuric acid) was co-injected with all sample injections. For construction of a calibration line, two standards were used containing, respectively, 10 mM and 100 mM of formic acid as well as several other organic acids (lactate, acetic acid, propionic acid, isobutyric acid and butyric acid). First, 1 µl, 2 µl and 5 µl of the 10 mM standard solution were injected to get calibration points for 1 mM, 2 mM and 5 mM formate, respectively. In addition, 1 µl, 2 µl, 5 µl and 10 µl of the 100 mM standard solution were injected to get calibration points for 10 mM, 20 mM, 50 mM and 100 mM formate, respectively. Calibration was validated using a commercial formate

standard containing 1,000 mg l$^{-1}$ ± 5 mg l$^{-1}$ (or 22.21 mM ± 0.11 mM) formate in water (44293 from Merck), of which 10 μl was injected. For sample measurement, 10 μl of sample was injected. No residual formate could be detected in the bioreactor samples. The biomass yield in grams CDW per mole formate was then calculated by dividing the CDW concentration (g l$^{-1}$) by the consumed formate concentration (mol l$^{-1}$).

## Whole-genome sequencing

*C. necator* genomes and plasmids were extracted from LB-grown cells for whole-genome sequencing using the NucleoSpin Microbial DNA Kit (Macherey-Nagel). Samples were sent for library preparation (Nextera, Ilumina) and sequencing at an Ilumina NovaSeq 6000 platform to obtain 150-bp paired-end reads (Novogene). Samples were paired, trimmed and assembled to the *C. necator* reference genome using the Geneious 8.1 software (Biomatters) or the Breseq pipeline[11]. Mutations (frequency above >60%) were identified based on comparative analysis with the parental strains.

## Max–min driving force analysis

Max–min driving force (MDF) analysis[12] was used to compare the thermodynamic feasibility of the CBB cycle and rGlyP for pyruvate formation from formate in an atmosphere of 10% $CO_2$ (v/v) (Supplementary Fig. 1). The Python packages equilibrator_api (v 0.4.7) and equilibrator_pathway (v 0.4.7) were used. Changes in Gibbs free energy of the reactions were estimated using the component contribution method[13]. Default values were used, metabolite and cofactor concentrations were constrained to the range 1 μM to 10 mM, pH was set to 7.5, ionic strength was assumed to be 0.25 M and magnesium concentration was 1 mM.

## Constraint-based metabolic modelling

The genome-scale metabolic model (GEM) of *C. necator*, RehMBEL1391_sbml_L3V1, was retrieved in SBML format level 3 version 1 from the public repository provided in ref. 6. Model simulations were performed using COBRApy 0.24.0 (ref. 14) and Python 3.9.

Flux balance analysis (FBA) was implemented to simulate formate assimilation for both the CBB cycle and the rGlyP. To simulate formate assimilation through the CBB cycle, the glycine cleavage system reaction ('GLYAMT') was set as non-reversible to prevent any flux through the rGlyP. To support formate assimilation through rGlyP, two additional reactions were added to the GEM: formate tetrahydrofolate ligase ('Ftl') and methenyltetrahydrofolate cyclohydrolase ('Fch'). In addition, the GLYAMT reaction was set as reversible, and the flux through the ribulose-bisphosphate carboxylase reaction ('RBPC') was set to 0 to prevent flux through the CBB cycle.

The ATP hydrolysis part of the biomass synthesis reaction was amended with an $H_2O$ molecule and $H^+$ to balance the reaction. The growth-associated maintenance (GAM; 135 mmol gCDW$^{-1}$) and non-growth-associated maintenance (NGAM; 3.0 mmol gCDW$^{-1}$ h$^{-1}$) values used to run the simulations were retrieved from ref. 6.

More information on their fine-tuned GAM parameter can be found between lines 101 and 104 of the script called 'run_simulations.py' within the reported GitLab repository (https://github.com/m-jahn/genome-scale-models/blob/master/Ralstonia_eutropha/).

To calculate the theoretical biomass yield, we constrained the biomass reaction to the growth rate corresponding to 14 h doubling time, the growth rate used in the bioreactors (Fig. 2d). The formate uptake rate reaction ('EX_formate_e') was used as the objective function, and fluxes were computed for each scenario under the specified conditions. The ratio between growth rate (h$^{-1}$) and the maximum predicted formate uptake rate (mmol g$^{-1}$ CDW h$^{-1}$) was used to calculate the biomass yields in g CDW mol$^{-1}$ formate.

## Proteomic analysis

*C. necator* CBB, CRG4 and CRG6 strains were pre-cultured in 4 ml M9 minimal medium supplemented with 80 mM formate and 100 mM HCO$_3$ in 15-ml glass tubes. Then, 1 ml of cells in the late exponential phase was collected and washed three times in M9 medium without a carbon source. From here, 50 ml of M9 minimal medium supplemented with 80 mM formate and 100 mM HCO$_3$ was inoculated to a starting OD$_{600}$ of 0.01 in 250-ml non-baffled shake flasks. Cultures were incubated in an Infors Minitron at 30 °C in an atmosphere of 10% $CO_2$ (v/v) at 200 rpm. This flask-adapted culture was then used to inoculate a second flask culture (the third consecutive formate cultivation) in the same way. Cells from 3 biological replicates were collected in the mid-log phase (OD$_{600}$ = 0.3–0.5) and washed twice with phosphate buffer (12 mM phosphate buffer, 2.7 mM KCl, 137 mM NaCl, pH = 7.4). Cell pellets corresponding to 1 ml of OD$_{600}$ = 3 were flash-frozen in liquid nitrogen and stored at −70 °C until further use. Cell lysis and protein solubilization were conducted as previously reported. In brief, cells were incubated 15 min at 90 °C in 2% sodium lauroyl sarcosinate (SLS) and 100 mM ammonium bicarbonate and then sonicated for 15 s (Vial Tweeter, Hielscher). Soluble proteins were reduced via incubation with 5 mM Tris (2-carboxy-ethyl) phosphine (TCEP) at 90 °C for 15 min, followed by alkylation with 10 mM iodoacetamide for 15 min at 25 °C. Protein concentrations were quantified via a bicinchoninic acid assay (BCA) protein assay kit (Thermo Fisher Scientific). Then, 50 μg of protein was digested with 1 μg trypsin (Promega) in 0.25% SLS (diluted with 100 mM ammonium bicarbonate) overnight at 30 °C. Following SLS removal via centrifugation, trifluoroacetic acid (TFA) was added to a final concentration of 1.5% and samples were incubated at room temperature for 10 min. The supernatant was purified using C18 Micro Spin Columns (Harvard Apparatus) according to the manufacturer's instructions, dried and resuspended in 0.1% TFA. Peptide mixtures were then analysed using liquid chromatography–mass spectrometry carried out on an Exploris 480 instrument connected to an Ultimate 3000 RSLC nano with a Proflow upgrade and a nanospray flex ion source (all Thermo Scientific). Peptide separation was performed on a reverse-phase HPLC column (75 μm × 42 cm) packed in-house with C18 resin (2.4 μm, Dr. Maisch). The following separating gradient was used: 94% solvent A (0.15% formic acid) and 6% solvent B (99.85% acetonitrile, 0.15% formic acid) to 25% solvent B over 95 min and to 35% B for an additional 25 min at a flow rate of 300 nl min$^{-1}$. The data-independent acquisition-mass spectrometry (DIA-MS) acquisition method was adapted from ref. 15. In short, the spray voltage was set to 2.0 kV, the funnel radio frequency (RF) level at 45 and the heated capillary temperature at 275 °C. For DIA experiments, full MS resolutions were set to 120,000 at m/z 200, and the full MS automatic gain control (AGC) target was 300% with an injection time (IT) of 50 ms. The mass range was set to 350–1,400. The AGC target value for fragment spectra was set at 3,000%. A total of 49 windows of 15 Da were used with an overlap of 1 Da. The resolution was set to 15,000 and the IT to 22 ms. Stepped higher-energy collisional dissociation (HCD) collision energy of 25%, 27.5% and 30% was used. MS1 data were acquired in profile, MS2 DIA data in centroid mode.

Analysis of DIA data was performed using DIA-NN version 1.8 (ref. 16), using the UniProt protein database from *C. necator* H16 and added sequences for formate–tetrahydrofolate (THF) ligase (*ftfL*, UniProt: Q83WS0), 5,10-methenyl-THF cyclohydrolase (*fchA*, UniProt: Q49135) and 5,10-methylene-THF dehydrogenase (*mtdA*, UniProt: P55818) from *Methylorubrum extorquens AM1*, RK2 plasmid replication protein (trfA, UniProt: P07676), pBBR1 replication protein (pSEVA331 derived AA sequence), aminoglycoside 3′-phosphotransferase (aphA1, Uniprot: P00551) and chloramphenicol acetyltransferase (cat, Uniprot: P62580). Full tryptic digest was allowed with three missed cleavage sites, and oxidized methionines and carbamidomethylated cysteines. Match between runs and remove likely interferences were enabled. The neural network classifier was set to the single-pass mode, and protein inference was based on genes. Quantification strategy was set to any LC (high accuracy). Cross-run normalization was set to retention time (RT) dependent. Library generation was set to smart profiling. DIA-NN

outputs were further evaluated using a SafeQuant version modified to process DIA-NN outputs[17].

The different metabolic groups were defined as follows. 'rGlyP-C1' is composed of FtfL, Fch and MtdA, 'rGlyP-C2' of GcvT1HP and 'rGlyP-C3' of SdaA and GlyA. The 'plasmid backbone' proteome is composed of TrfA, AphA1, pBBR Rep and Cat. 'CBB cycle' proteins contain when applicable CbbL1, CbbL2, CbbS, CbbS2, CfxP, CbxXC, CbbYC, CbbAC, CbbAP, Fbp2, Fbp3, Rpe1, Rpe2, CbxXP, CbbTC, CbbTP, CbbZC, CbbZP, CbbKC, CbbKP, CbbGC and CbbGP. The 'hydrogen utilization' proteome contains the proteome fractions of HoxA, HoxB, HoxC, HoxF, HoxG, HoxH, HoxI, HoxK, HoxL, HoxM, HoxN, HoxO, HoxQ, HoxR, HoxU, HoxV, HoxW, HoxY, HoxZ, HypA, HypB, HypB2, HypC, HypD, HypE, HypF1, HypF2 and HypX. 'Formate utilization' is composed of the proteins FdsD, FdsA, FdsB, FdsG, FdsR, FdoI, FdoH, FdoG, FdhA1, FdhA2, FdhB1, FdhC, FdhD, FdhD1, FdhD2, FdhE, FdwA, FdwB and CbbB. rpsA, rpsB, rpsC, rpsD, rpsE, rpsF, rpsG, rpsH, rpsI, rpsJ, rpsK, rpsL, rpsM, rpsN, rpsO, rpsP, rpsQ, rpsR, rpsS, rpsT, rpsU, rpsU, rplA, rplB, rplC, rplD, rplE, rplF, rplI, rplJ, rplK, rplL, rplM, rplN, rplO, rplP, rplQ, rplR, rplS, rplT, rplU, rplV, rplW, rplX, rplY, rpmA, rpmB, rpmC, rpmD, rpmE2, rpmF, rpmG, rpmH, rpmI and rpmJ make up the ribosome group. All other detected proteins (~3,500) make up the 'other' category.

## Figure preparation
Perseus 1.5.1.6 was used to analyse and plot proteomics data. Biomass yield data were plotted and checked for significance in GraphPad Prism 10.1.0 using a two-tailed, unpaired *t*-test. Adobe Illustrator 2020 was used to make the illustrations for the workflow, the transposon insertion sites and the formate dehydrogenase inhibition trials, as well as to prepare and export the final figures.

## Reporting summary
Further information on research design is available in the Nature Portfolio Reporting Summary linked to this article.

## Data availability
Data supporting the findings of this study are found in the Article and Supplementary Information. Sequence data of plasmids constructed and/or used in this study were made available via EDMOND (https://doi.org/10.17617/3.FSBOQE). Whole-genome sequencing data have been deposited in NCBI Sequence Read Archive (SRA) under BioProject number PRJNA1209586. The mass spectrometry proteomics data have been deposited in ProteomeXchange Consortium via the PRIDE partner repository with dataset identifier PXD059545. Source data are provided with this paper. Further information, bacterial strains and materials related to this study are available from the corresponding authors.

## Code availability
The GEM of *C. necator*-RehMBEL1391_sbml_L3V1, the code and the associated results obtained in this study are available via GitHub at https://github.com/benit005/Formate_assimilation_rGlyPvsCBB_Cupriavidus_necator.

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

## Acknowledgements
We dedicate this work to the memory of A. Bar-Even (1980–2020) who was involved in the design and the initial phase of this study. Above all, he had been our exceptional and dear mentor and colleague. We thank O. Lenz for providing plasmid pLO3 and advice on working with *C. necator*. In addition, we acknowledge P. Nikel for providing the pBAMD plasmids and W. Newell and J. Kahnt for experimental assistance. We also thank N. Paczia and P. Pfister for technical assistance with the bioreactor cultivations. We thank M. Jahn for input on the quantitative proteomic data analysis and G. Angelidou for help with bioinformatic analysis. Lastly, we thank A. Satanowski and S. Wenk for critical reading of the paper. This study was funded by the Max Planck Society (to B.D., E.O., H.S.-M., T.G., A.B.-E. and T.J.E.), the German Ministry of Education and Research via the Transformate grant (grant number 033RC023G to B.D., E.O. and A.B.-E.), an Add-On Fellowship for Interdisciplinary Life Science of the Joachim Herz Foundation and the Bosch Research Foundation (to H.S.-M.) and the Dutch Science Organization (NWO) with a Veni grant (grant number VI.Veni.192.156 to N.J.C.) and a Gravitation Project BaSyC (grant number 024.003.019 to S.Y. and N.J.C.).

## Author contributions

A.B.-E., B.D., E.O. and N.J.C. designed and conceived the study. A.B.-E., N.J.C. and T.J.E. supervised the project. B.D. performed cloning, strain engineering and NGS analysis. B.D., S.Y. and H.S.-M. performed experimental characterizations. T.G. and B.D. performed the proteomic analysis. S.B.-V. and H.S.-M. conducted the modelling. B.D., E.O., T.J.E. and N.J.C. wrote the paper. The other authors provided feedback and approved the paper.

## Funding

## Competing interests

The authors declare no competing interests.

## Additional information

Extended data is available for this paper at

Supplementary information The online version
contains supplementary material available at

Beau Dronsella or Nico J. Claassens.

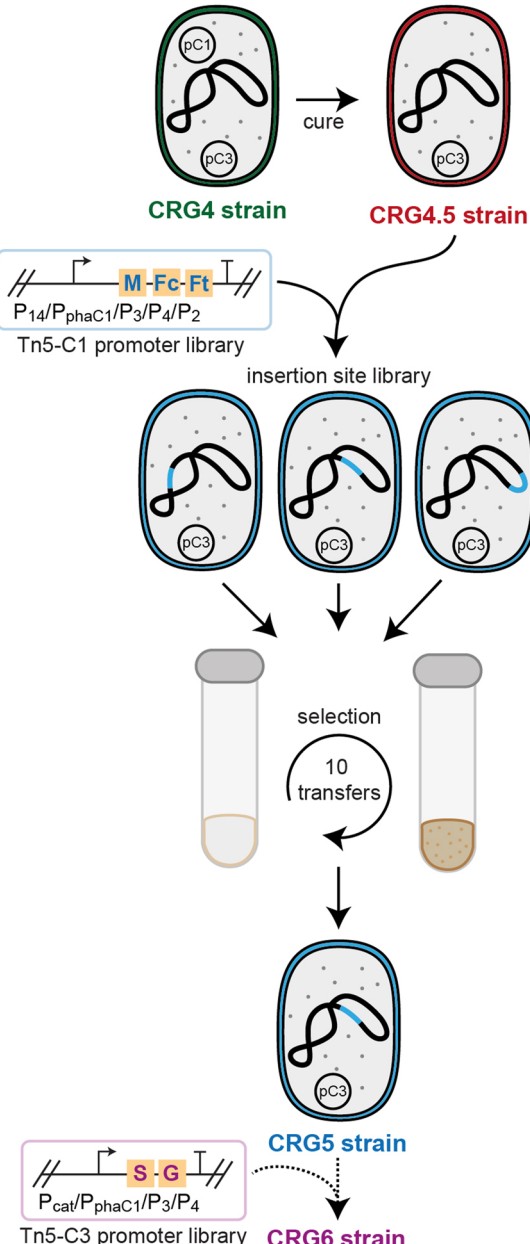

**Extended Data Fig. 1 | The genetic integration workflow employed in this study.** The two synthetic operons previously expressed from plasmids were genomically integrated via random Tn5-transposon integration with a library of different strength promoters in two consecutive rounds. The generated genetic libraries were grown on formate medium in 10 rounds to select for the fastest growing strains.

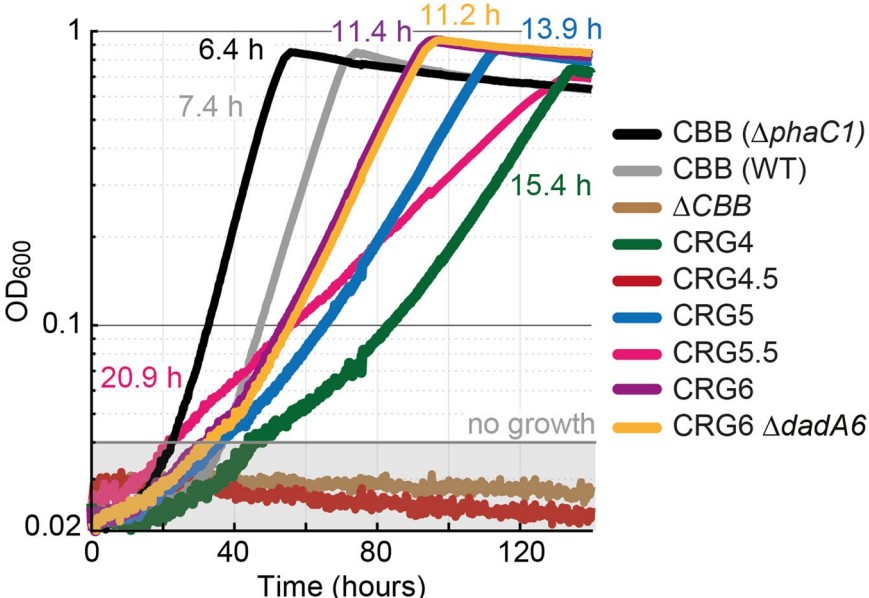

**Extended Data Fig. 2 | Overview of strains and their phenotypes in batch cultivation.** Growth of various *C. necator* strains used in this study on M9 minimal media supplemented with 80 mM formate and 100 mM bicarbonate with 10 % $CO_2$ in the headspace. The doubling time in hours (h) of the strains is presented in the designated strain color. Curves depict the mean of 2–3 technical replicates and are representative of 3 experiments conducted in the same conditions to ensure reproducibility. Less than one doubling of the bacterial strains was considered no growth and indicated as such via an overlayed shaded box.

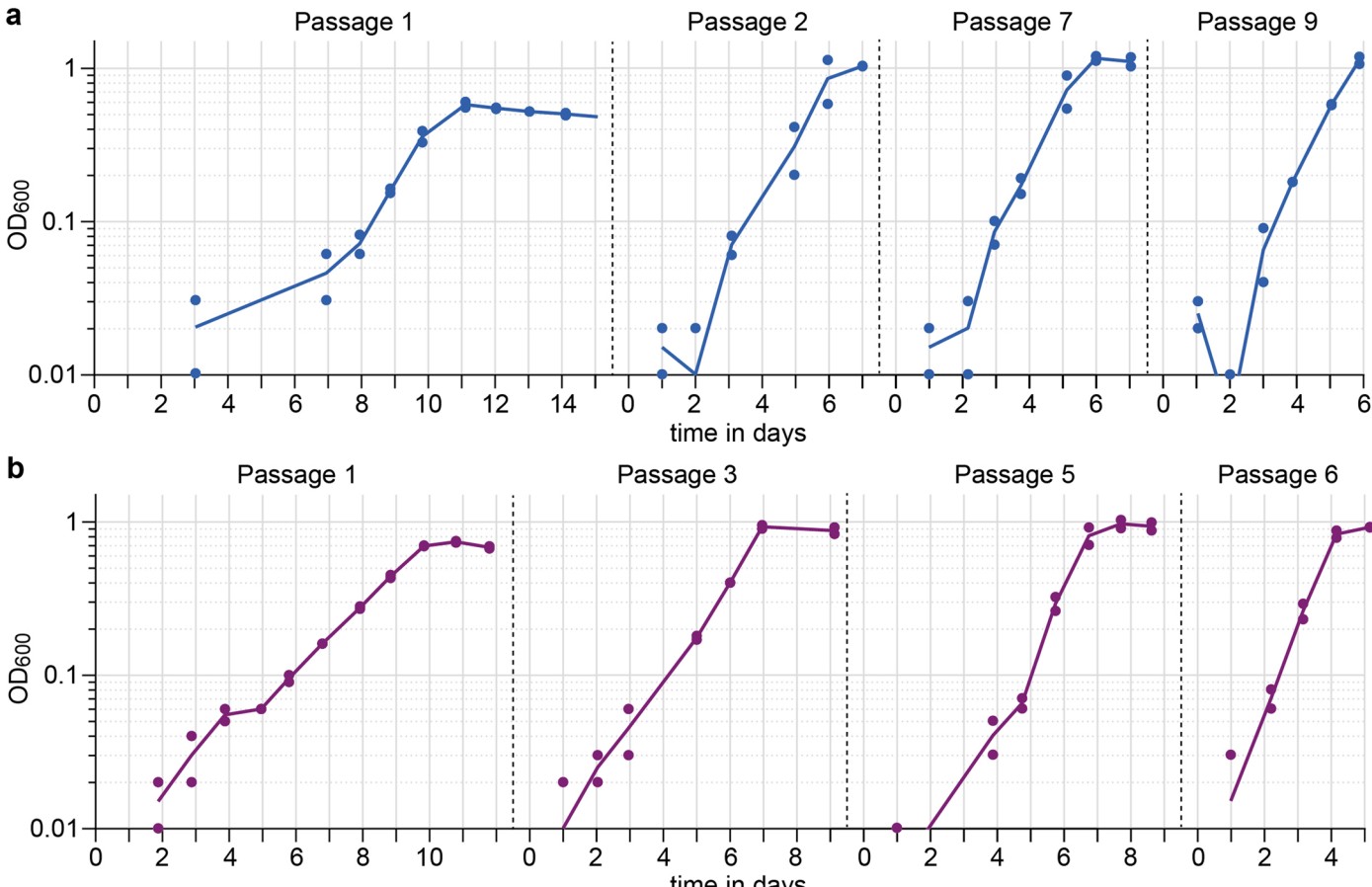

**Extended Data Fig. 3 | Creation of CRG5 and CRG6 strains.** Liquid selection test tube experiment on M9, 80 mM formate, 100 mM $HCO_3$ minimal media in 10% $CO_2$ for creation of the CRG5 $gP_{14}$-C1 (**a**) and CRG6 $gP_3$-C3 strains (**b**). Following transposon integration of module 1 or module 3 of the rGlyP and upon reaching early stationary phase cells were re-inoculated into fresh formate media. Dots represent the measured glass tube $OD_{600}$ values of two biological replicates per promoter operon transposon insertion. The lines correspond to their averages. 4 passages are depicted and are indicative of the improved growth over the 10 passages.

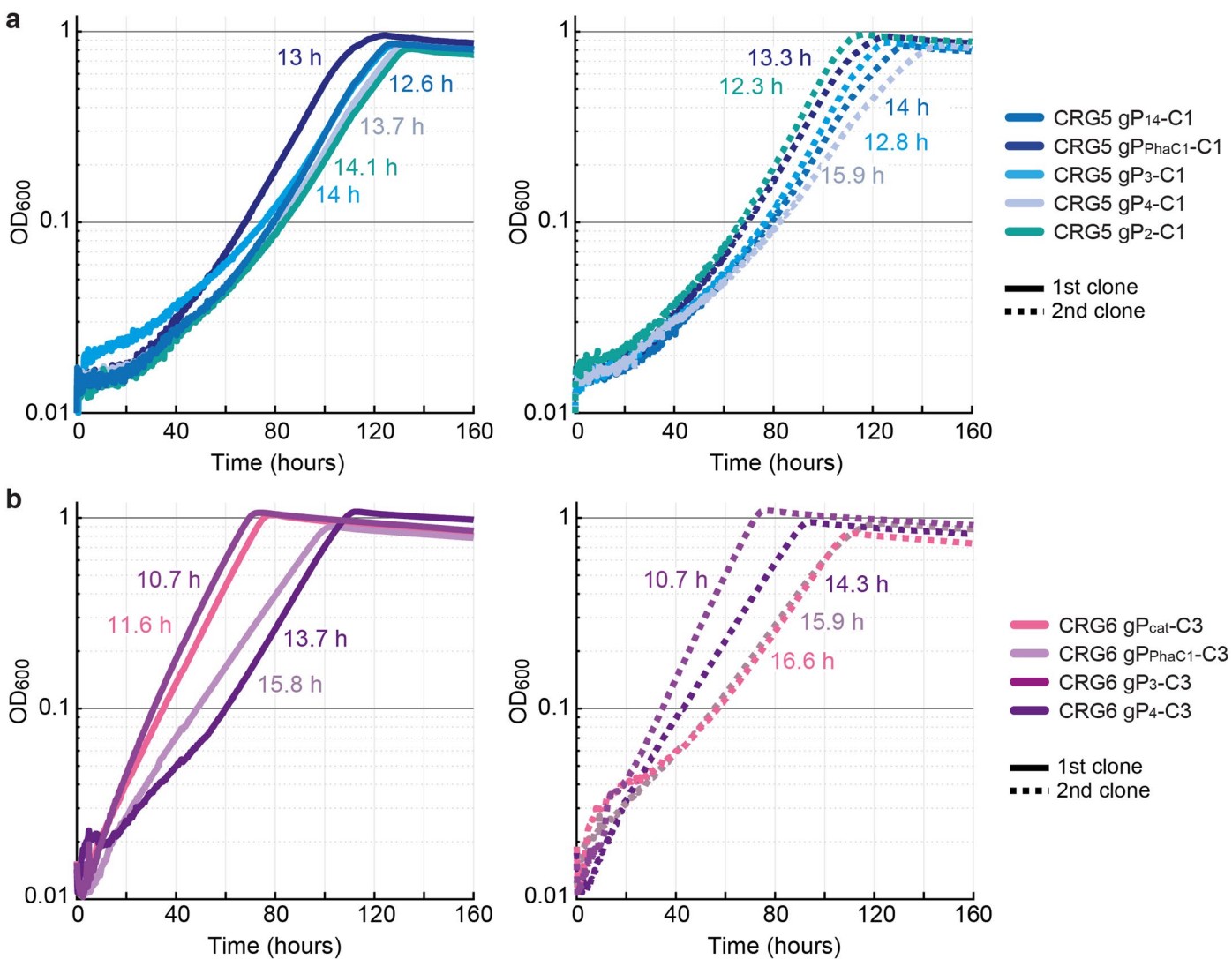

**Extended Data Fig. 4 | Characterization of CRG5 and CRG6 lineages with different promoters.** Growth profiles of isolated single clones after the liquid selection experiment of (**a**) CRG5 and (**b**) CRG6 strains with different promoter C1 and C3-module transposon insertions on M9, 80 mM formate, 100 mM $HCO_3$ minimal media in 10% $CO_2$. Promoters $P_{14}$, $P_{PhaC1}$, $P_3$, $P_4$ and $P_2$ were used to control the C1 operon, while the promoters $P_{cat}$, $P_{PhaC1}$, $P_3$, $P_4$ were used to control the C3 operon. Two individually transposed lineages are shown per promotor, solid lines indicate the first clone and the dotted line the second. Doubling time in hours is given in the respective strain color beside the growth curve. Curves are means of 3 or 4 technical replicates for CRG5 and CRG6 respectively and representative of 3 experiments conducted in the same conditions. 1st clone strains CRG5 $gP_{14}$-C1 and CRG6 $gP_3$-C3 were the focus of this study and have the corresponding color shade associated with them in the main text figures.

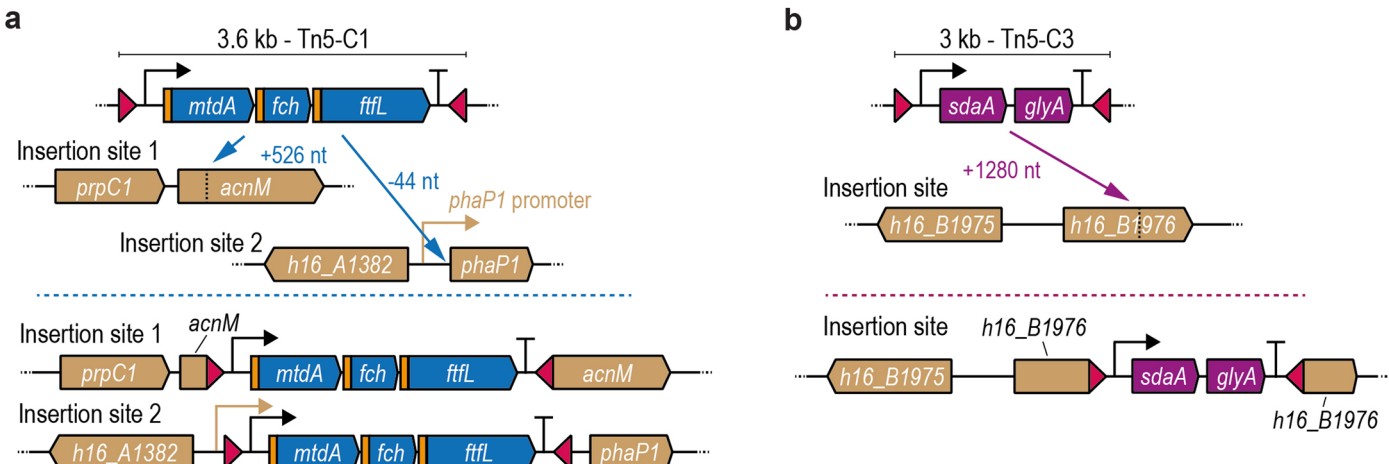

**Extended Data Fig. 5 | Transposon insertion loci of the genomic rGlyP modules in this study.** (**a**) the CRG5 lineage carries two insertions of the Tn5-P$_{14}$-C1 construct. One in the *acnM* gene and one upstream of the *phaP1* gene. (**b**) the transposon expressing the P$_3$-C3-module inserted in the gene *H16_B1976* in strain CRG6.

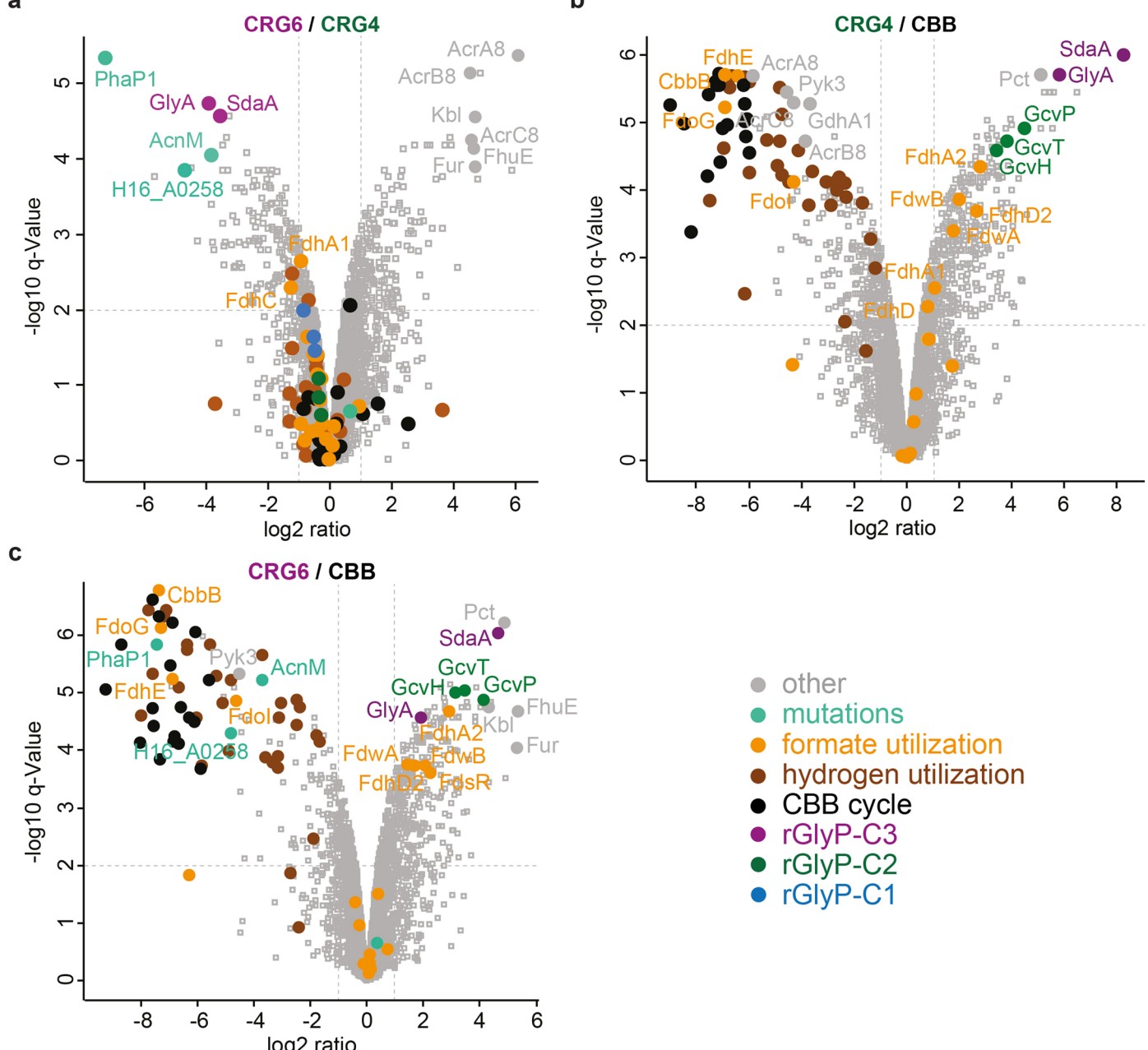

**Extended Data Fig. 6 | Proteomic comparison of rGlyP and CBB cycle strains.** Global changes in the proteomes of (**a**) CRG4 and (**b**) CRG6 compared to the CBB cycle strain and of CRG6 to CRG4 (**c**). Log2-fold changes in protein intensity are plotted against significance (-log 10 q-Values). Relevant proteins are labeled, shown as a filled-out circles and colored according to their metabolic task. rGlyP proteins associated with module 1 are shown in blue, module 2 in green and module 3 in purple. CBB cycle and hydrogen utilization proteome are depicted in black and brown respectively. Different formate utilization proteins are highlighted in orange. Loci hit by transposon insertions and the frequently observed mutations as in H16_A0258 are shown in turquois. The most up and downregulated proteins if not already categorized, are colored in grey. All other proteins are not labeled, depicted as empty squares and colored grey.

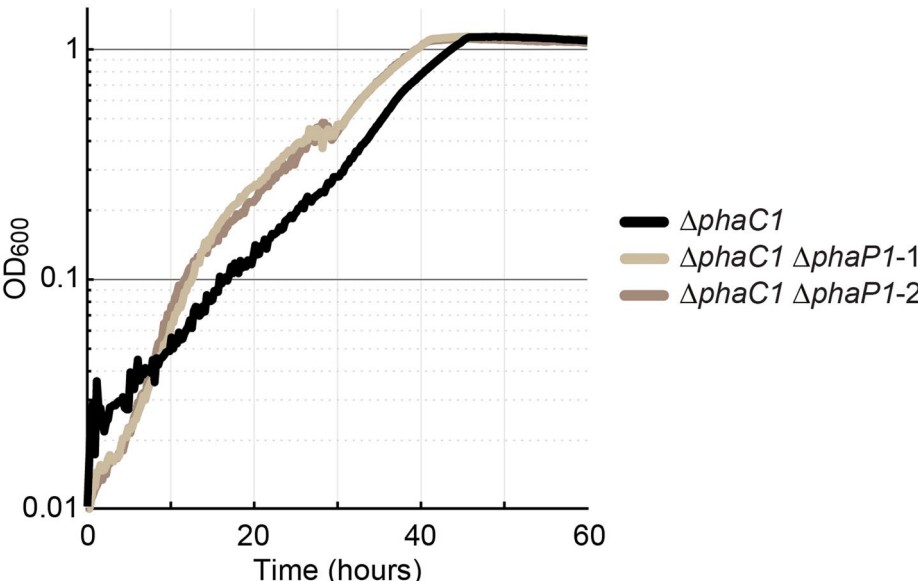

**Extended Data Fig. 7 | Effect of *phaP1* deletion on growth of the CBB cycle strain.** Strains were cultivated in M9 media supplemented with 80 mM formate and 100 mM $HCO_3$ with 10 % $CO_2$ in the headspace in a 96-well plate reader. Two biological clones of the Δ*phaC1* Δ*phaP1* strain are shown. Curves depict the mean of 4 technical replicates.

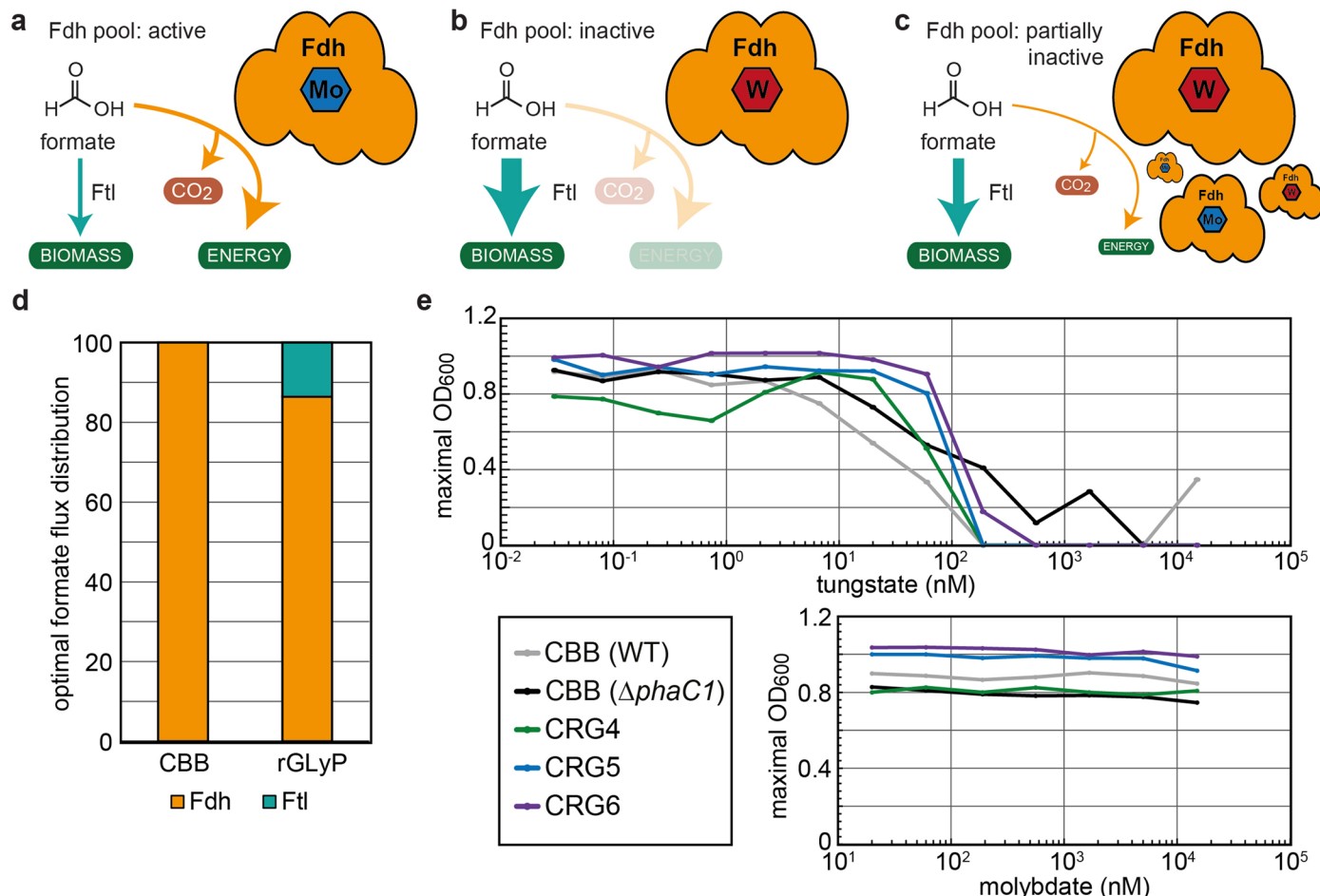

**Extended Data Fig. 8 | Effect of sFDH inhibition on biomass yield (OD600).** Depending on the presence of the activating or inhibiting metals molybdenum and tungsten, the sFDH pool can either be fully active (**a**), inactive (**b**), or partially inactive (**c**), when both are present in varying concentrations. (**d**) FBA-based prediction of the required oxidative and assimilatory fluxes for Calvin cycle (CBB) and reductive glycine pathway (rGlyP) at 14 h doubling time. (**e**), *C. necator*

*H16* CBB (WT), CBB (Δ*phaC1*), CRG4, CRG5 and CRG6 strains were grown on M9, 80 mM formate, 100 mM HCO$_3$ minimal media in 96-well plates. Cells were either supplemented with varying concentrations of sodium tungstate to inhibit sFDH or sodium molybdate to potentially further increase sFDH activity with the appropriate metal ion.

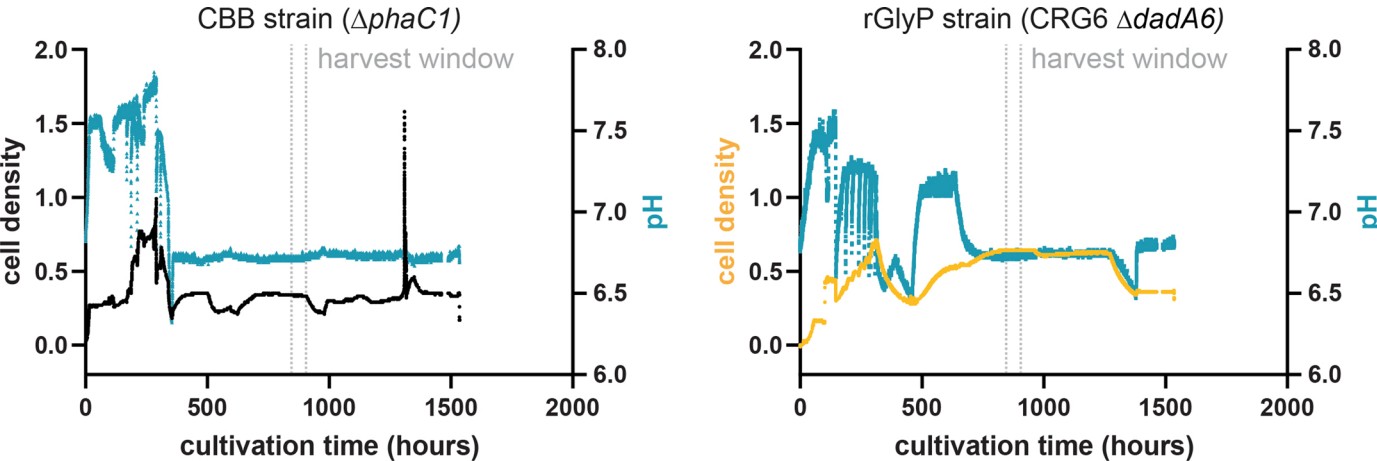

**Extended Data Fig. 9 | Chemostat cultivations of *C. necator* Δ*phaC1* and CRG6 Δ*dadA6* on formate.** Strains were cultivated in a carbon limited chemostat at a dilution rate of 0.05 h⁻¹ with a 80 mM formic acid feed in 10 % CO₂ (at 2.5 vvm). Process data from the entire Δ*phaC1* and CRG6 Δ*dadA6* cultivations are shown. Infrared based cell density data and pH values are depicted. The 3 day harvest window during which samples were taken for cell dry weight analysis is indicated in grey. The dilution rate was set to 0.087 h⁻¹ equal to 8 h doubling time after 1000 h of cultivation and to 0.1 h⁻¹ after ~1300 h of cultivation, after which the cells washed out.

Nico Claassens

# Reporting Summary

## Statistics

For all statistical analyses, confirm that the following items are present in the figure legend, table legend, main text, or Methods section.

| n/a | Confirmed | |
|---|---|---|
| ☐ | ☒ | The exact sample size (*n*) for each experimental group/condition, given as a discrete number and unit of measurement |
| ☐ | ☒ | A statement on whether measurements were taken from distinct samples or whether the same sample was measured repeatedly |
| ☐ | ☒ | The statistical test(s) used AND whether they are one- or two-sided *Only common tests should be described solely by name; describe more complex techniques in the Methods section.* |
| ☒ | ☐ | A description of all covariates tested |
| ☒ | ☐ | A description of any assumptions or corrections, such as tests of normality and adjustment for multiple comparisons |
| ☐ | ☒ | A full description of the statistical parameters including central tendency (e.g. means) or other basic estimates (e.g. regression coefficient) AND variation (e.g. standard deviation) or associated estimates of uncertainty (e.g. confidence intervals) |
| ☐ | ☒ | For null hypothesis testing, the test statistic (e.g. *F*, *t*, *r*) with confidence intervals, effect sizes, degrees of freedom and *P* value noted *Give P values as exact values whenever suitable.* |
| ☒ | ☐ | For Bayesian analysis, information on the choice of priors and Markov chain Monte Carlo settings |
| ☒ | ☐ | For hierarchical and complex designs, identification of the appropriate level for tests and full reporting of outcomes |
| ☒ | ☐ | Estimates of effect sizes (e.g. Cohen's *d*, Pearson's *r*), indicating how they were calculated |

*Our web collection on statistics for biologists contains articles on many of the points above.*

## Software and code

Policy information about availability of computer code

| Data collection | No special software was used. |
|---|---|
| Data analysis | The following software was used: MatLab R2019a, equilibrator-api 0.2.5, Geneious 8.1, Breseq (v0.36.1), SnapGene 8.0.1, Graph Pad Prism 10.1.0, Adobe Illustrator 2020, Perseus 1.5.1.6 |

For manuscripts utilizing custom algorithms or software that are central to the research but not yet described in published literature, software must be made available to editors and reviewers. We strongly encourage code deposition in a community repository (e.g. GitHub). See the Nature Portfolio guidelines for submitting code & software for further information.

## Data

Policy information about availability of data

All manuscripts must include a data availability statement. This statement should provide the following information, where applicable:

- Accession codes, unique identifiers, or web links for publicly available datasets
- A description of any restrictions on data availability
- For clinical datasets or third party data, please ensure that the statement adheres to our policy

Data supporting the findings of this study are found in the paper, the extended data figures, the supplementary information, or the source data files. Sequence data of plasmids constructed and/or used in this study was made available on EDMOND (https://doi.org/10.17617/3.FSBOQE). Whole-genome sequencing data were deposited on the NCBI sequencing read archive (SRA) under the BioProject number PRJNA1209586. The mass spectrometry proteomics data have been deposited

# Research involving human participants, their data, or biological material

Policy information about studies with <u>human participants or human data</u>. See also policy information about <u>sex, gender (identity/presentation), and sexual orientation</u> and <u>race, ethnicity and racism</u>.

| | |
|---|---|
| Reporting on sex and gender | N/A |
| Reporting on race, ethnicity, or other socially relevant groupings | N/A |
| Population characteristics | N/A |
| Recruitment | N/A |
| Ethics oversight | N/A |

Note that full information on the approval of the study protocol must also be provided in the manuscript.

# Field-specific reporting

Please select the one below that is the best fit for your research. If you are not sure, read the appropriate sections before making your selection.

☒ Life sciences ☐ Behavioural & social sciences ☐ Ecological, evolutionary & environmental sciences

For a reference copy of the document with all sections, see nature.com/documents/nr-reporting-summary-flat.pdf

# Life sciences study design

All studies must disclose on these points even when the disclosure is negative.

| | |
|---|---|
| Sample size | Sample sizes for proteomics were in biological triplicates, which is a generally accepted number for similar microbial experiments. Sample size of growth experiments were 2-3 technical replicates with the experiments being repeated at least 3 times to ensure reproducibility, which reflects the state of the art in the field (10.1038/s41467-024-53762-9). The chemostat biomass yield experiment was conducted in the same way as an established reference in the field (10.1111/1751-7915.12149), making results more comparable. We harvested over the course of 3 days a total of 9 samples to assure technical replication. We further were able to benchmark our data to the aforementioned reference, have several OD600 based growth experiments showing similar trends in biomass formation, and have preprinted flask yield data further confirming our results. |
| Data exclusions | No data were excluded. |
| Replication | Growth experiments in glass tubes and plate readers were repeated 3 times to ensure reproducibility. Biomass yields are derived from 9 technical replicates taken from the chemostat in steady-state over the course of 3 days. |
| Randomization | Biological replicates were chosen at random from plate, otherwise no further randomization was performed. |
| Blinding | Blinding was not performed, as this study was conducted using a single microorganism in a controlled laboratory environment, with appropriate controls to avoid bias. |

# Reporting for specific materials, systems and methods

We require information from authors about some types of materials, experimental systems and methods used in many studies. Here, indicate whether each material, system or method listed is relevant to your study. If you are not sure if a list item applies to your research, read the appropriate section before selecting a response.

## Materials & experimental systems

| n/a | Involved in the study |
|-----|----------------------|
| ☒ ☐ | Antibodies |
| ☒ ☐ | Eukaryotic cell lines |
| ☒ ☐ | Palaeontology and archaeology |
| ☒ ☐ | Animals and other organisms |
| ☒ ☐ | Clinical data |
| ☒ ☐ | Dual use research of concern |
| ☒ ☐ | Plants |

## Methods

| n/a | Involved in the study |
|-----|----------------------|
| ☒ ☐ | ChIP-seq |
| ☒ ☐ | Flow cytometry |
| ☒ ☐ | MRI-based neuroimaging |

## Plants

| Seed stocks | N/A |
|-------------|-----|
| Novel plant genotypes | N/A |
| Authentication | N/A |

