## [Peer Review File · Nature Microbiology]

One-carbon fixation via the reductive glycine pathway exceeds yield of the Calvin cycle in bacteria

Corresponding Author: Dr Nico (J) Claassens

Version 0:

Reviewer comments:

Reviewer #1

(Remarks to the Author)

Dronsella et al. demonstrated an engineered strain of *Cupriavidus necator*, utilizing a synthetic, more energy-efficient reductive glycine pathway (rGlyP) rather than the native Calvin-Benson-Bassham (CBB) cycle, resulting in higher biomass yield than the wild type while assimilating formic acid. Building on their previous study (ref 21), which used a plasmid-based approach to introduce the rGlyP pathway, the present manuscript employed a Tn5-based transposon to avoid the plasmid burden. While this result is encouraging, this reviewer has some concerns regarding the present manuscript, which are detailed below.

1. This work uses the same rGlyP as the previous publication (ref. 21), but the previous attempt did not generate a better biomass yield than the WT strain. The authors explained this difference by plasmid burden. However, there is no convincing data to support his argument.
 2. To our understanding, the limiting enzyme in the CBB pathway, Rubisco, is a slow CO₂ fixing enzyme. rGlyP strains theoretically should grow faster than CBB strains, but the result demonstrates the opposite. Please explain.
 3. Although biomass yield and growth rate are two different parameters, they roughly correlate with each other in a non-linear fashion. However, the rGlyP shows opposite effects on the biomass and the growth rate. Please explain.
 4. The authors reported that their engineered strain reached a 14% higher biomass yield than the CBB cycle-utilizing wild type. However, Calvey et al. (2023) reported that the wild type exhibited 1.15x to 2.18x faster maximum growth rates and 10%–34% greater maximum optical density at OD₆₀₀ after serial passages. It should be noted that the engineered strain and wild-type control in this manuscript have not been compared on the same basis, as the engineered strain has been passaged while the wild-type strain has not.
 5. Can the authors prove that the biomass yield increase is a consequence of rGlyP instead of serial passages? Perhaps GRC4 strain after a few passages can also reach higher biomass yield.
- Points No.4 & 5 cast some doubts about the superiority of the rGlyP pathway. Please explain.
6. The authors claimed theoretical maximum biomass yield are derived from constraint-based metabolic modelling. Can you elaborate more about GAM and NGAM, and how to get these values?

Reviewer #2

(Remarks to the Author)

Dronsella et al, Engineered synthetic one-carbon fixation exceeds yield of the Calvin Cycle – Review Nature Microbiology

In this manuscript the authors use transposable element genomic integration of 6 enzymes which were previously harbored on plasmids to show increased efficiency of the rGly pathway to formate fixation to biomass in *Cupriavidus necator*. Screening after transposable element integration and plasmid curing was performed by analyzing growth rates to identify fast growing strains fed by formate and in a CO₂ enriched environment. The integrative elements cause knockout of redundant genes and also allow high transcription rates of the target genes that, through proteomics, matched amino acid amounts of those achieved from plasmid expression of the target pathway. The authors show, for the first time, that this engineered pathway to carbon fixation can outcompete the native fixation of CO₂ to biomass from the ATP derived from formate oxidation (to CO₂) in this organism by its inherent CBB cycle.

The work is of high relevance to the microbial engineering community and should of course be published. There are some small changes which would make the manuscript more accessible to the reader which I would propose be made prior to

acceptance.

1. CBB is abbreviated in its first appearance in the introduction, I believe the abstract does not count as a first appearance for the full words and should be repeated here the first time.
2. A figure illustrating formate use through the CBB cycle should be included with the synthetic carbon fixation cycle so the reader can instantly see the differences and intuitively understand the advantage of the rGly pathway.
3. The introduction mentions anaerobic and aerobic rGly, but does not specify which the engineered one here represents and why it matters.
4. The introduction could benefit from being somewhat more didactic in topic presentation: Something along the lines of "Formate oxidation generates ATP, non-oxidized formate enters the cycle to make glycine and eventually pyruvate. In the CBB cycle, formate oxidation yields ATP, and the CO₂ is then captured into the CBB." The authors should write these helper sentences in the introduction as if the reader has not read every other paper on this topic, in order to make it a more valuable standalone literature reference.
5. Rubisco, would be more appropriately written as RuBisCO and requires at least one full acronym description in its first mention.
6. Line 67-68 – mention of the organism's use as agricultural protein feed must be elaborated. Is this microbe GRAS for consumption? Does it make large amounts of biomass per unit volume time? What makes it special, other than it can produce biomass from formate?
7. *C. necator* has changed names several times, some mention of this in the introduction would be important.

Scientific questions for the authors:

1. Why is a CO₂ atmosphere needed? Should not the CO₂ formed from formate oxidation result in biomass formation?
2. From a space-time yield of biomass per liter, how does formate consumption in the newly engineered strain compare to photolithoautotrophic rates in photosynthetic organisms like green algae or cyanobacteria? Would FDH expression in an alga and formate feeding result in faster heterotrophic growth, yes it requires more ATP per cycle, but what would be the benefits of *C. necator* over other CBB containing hosts? This should be discussed in the main text.

Reviewer #3

(Remarks to the Author)

In this study, the authors continued their work from the previous study (<https://doi.org/10.1016/j.ymben.2020.08.004>) to improve the strength of synthetic glycine pathway in *Cupriavidus necator*, an aerobic formatotroph. By using Tn5 transposon machinery, modules of rGly pathway were suitably integrated and expressed in *Cupriavidus necator* that enhanced the efficiency of formate use in biomass formation compared to the prior strain. The authors have conducted fundamental experiments to comprehend all strains generated from engineering process. The authors supplied proteomic data to gain a thorough understanding of the distribution of the synthetic rGly pathway. However, the authors should improve the discussion rather than only describe the results. Moreover, below are some comments to help the manuscript to be understood better.

Comments:

- 1) Please provide the information on various promoters used in this study: P14, PphaC, P3, P4, P2.
- 2) Please describe how the recombinant plasmids pC1 and pC3 were cured from the engineered cells.
- 3) Please describe briefly the CRG4 strain at the beginning of the Result section to be understood better.
- 4) Formatotrophic growth in this study needs both formate and carbon dioxide (or bicarbonate) as carbon sources. Thus, calculation of the biomass yield based on gram cell dry weight per mole formate might be not adequate. What about the difference between consumption and generation of carbon dioxide (or bicarbonate) by the cell ?
- 5) Is there any evidence for the inhibition of molybdenum-dependent formate dehydrogenase by addition of tungsten ?
- 6) It would be great if the authors compare the efficiency of synthetic reductive glycine pathway assembled in *Cupriavidus necator* and in other bacterial hosts from the literatures.
- 7) The authors claimed the more efficient formate-to-biomass process via implementation of the synthetic reductive glycine pathway; however, the doubling time of the rGly-utilizing cell was lower than the CBB-utilizing cell. Is there any considerable strategy to improve the doubling time of rGly-utilizing cells?
- 8) As polyhydroxybutyrate (PHB) is one of the practical products of *Cupriavidus necator*, efficiently production of PHB from C1 sources like formate and CO₂ using this host strain would be a wonderful strategy, which would bring economic benefit. However, the integration of module 1 of the rGly pathway at the phasin position might negatively affect the production of polyhydroxybutyrate, what should you deal with this situation?

Decision Letter:

9th February 2023

Dear Dr Claassens,

Thank you for your patience while your manuscript "Engineered synthetic one-carbon fixation exceeds yield of the Calvin Cycle" was under peer review at Nature Microbiology. It has now been seen by our referees, whose expertise and comments you will find at the end of this email. In the light of their advice, we have decided that we cannot offer to publish your manuscript in Nature Microbiology, at least in its present form.

From the reports, you will see that while they find your work of some potential interest, the referees raise concerns about the advance your findings represent over earlier work--especially in relation to your past paper detailing the use of rGlyP--and the strength of the novel conclusions that can be drawn at this stage. Reviewer #1, and to some extent Reviewer #3, raised a number of concerns related to technical issues and missing data. In particular, Reviewer #1 was not convinced that the high yield was as a result of rGlyP or simply serial passages. At this point, it is clear that a substantial amount of additional experimental work and explanation is needed, and it is not clear whether your conclusions will continue to be impressive in light of this additional work. While we would not rule out consideration of a revised manuscript that fully addresses the Reviewer's concerns with additional experimental work, at this point the criticisms are sufficiently important as to preclude further consideration of your work in Nature Microbiology and suggest that your best option is to submit the manuscript in its current form to another journal.

However, if you do feel that you would be able to include additional work to address these points, we would be willing to consider an appeal, although please note that we would reassess novelty with respect to existing literature at the time of appeal and would be unlikely to trouble the referee again unless we felt that their concerns had been satisfied in full. In the case of a successful appeal and eventual publication, the received date would be that of the revised paper.

I am sorry that we cannot be more positive on this occasion, but hope that you find the referees' comments helpful when deciding how to proceed.

Yours sincerely,

Reviewer Expertise:

Referee #1: metabolic engineering, systems biology, synthetic biology, carbon fixation

Referee #2: metabolic engineering, systems biology, synthetic biology, carbon fixation

Referee #3: metabolic engineering

Reviewers Comments:

Reviewer #1 (Remarks to the Author):

Dronsella et al. demonstrated an engineered strain of *Cupriavidus necator*, utilizing a synthetic, more energy-efficient reductive glycine pathway (rGlyP) rather than the native Calvin-Benson-Bassham (CBB) cycle, resulting in higher biomass yield than the wild type while assimilating formic acid. Building on their previous study (ref 21), which used a plasmid-based approach to introduce the rGlyP pathway, the present manuscript employed a Tn5-based transposon to avoid the plasmid burden. While this result is encouraging, this reviewer has some concerns regarding the present manuscript, which are detailed below.

1. This work uses the same rGlyP as the previous publication (ref. 21), but the previous attempt did not generate a better biomass yield than the WT strain. The authors explained this difference by plasmid burden. However, there is no convincing data to support his argument.
2. To our understanding, the limiting enzyme in the CBB pathway, Rubisco, is a slow CO₂ fixing enzyme. rGlyP strains theoretically should grow faster than CBB strains, but the result demonstrates the opposite. Please explain.
3. Although biomass yield and growth rate are two different parameters, they roughly correlate with each other in a non-linear fashion. However, the rGlyP shows opposite effects on the biomass and the growth rate. Please explain.
4. The authors reported that their engineered strain reached a 14% higher biomass yield than the CBB cycle-utilizing wild type. However, Calvey et al. (2023) reported that the wild type exhibited 1.15x to 2.18x faster maximum growth rates and 10%–34% greater maximum optical density at OD600 after serial passages. It should be noted that the engineered strain and wild-type control in this manuscript have not been compared on the same basis, as the engineered strain has been passaged while the wild-type strain has not.
5. Can the authors prove that the biomass yield increase is a consequence of rGlyP instead of serial passages? Perhaps GRC4 strain after a few passages can also reach higher biomass yield.
- Points No.4 & 5 cast some doubts about the superiority of the rGlyP pathway. Please explain.
6. The authors claimed theoretical maximum biomass yield are derived from constraint-based metabolic modelling. Can you

elaborate more about GAM and NGAM, and how to get these values?

Reviewer #2 (Remarks to the Author):

Dronsella et al, Engineered synthetic one-carbon fixation exceeds yield of the Calvin Cycle – Review Nature Microbiology

In this manuscript the authors use transposable element genomic integration of 6 enzymes which were previously harbored on plasmids to show increased efficiency of the rGly pathway to formate fixation to biomass in *Cupriavidus necator*. Screening after transposable element integration and plasmid curing was performed by analyzing growth rates to identify fast growing strains fed by formate and in a CO₂ enriched environment. The integrative elements cause knockout of redundant genes and also allow high transcription rates of the target genes that, through proteomics, matched amino acid amounts of those achieved from plasmid expression of the target pathway. The authors show, for the first time, that this engineered pathway to carbon fixation can outcompete the native fixation of CO₂ to biomass from the ATP derived from formate oxidation (to CO₂) in this organism by its inherent CBB cycle.

The work is of high relevance to the microbial engineering community and should of course be published. There are some small changes which would make the manuscript more accessible to the reader which I would propose be made prior to acceptance.

1. CBB is abbreviated in its first appearance in the introduction, I believe the abstract does not count as a first appearance for the full words and should be repeated here the first time.
2. A figure illustrating formate use through the CBB cycle should be included with the synthetic carbon fixation cycle so the reader can instantly see the differences and intuitively understand the advantage of the rGly pathway.
3. The introduction mentions anaerobic and aerobic rGly, but does not specify which the engineered one here represents and why it matters.
4. The introduction could benefit from being somewhat more didactic in topic presentation: Something along the lines of "Formate oxidation generates ATP, non-oxidized formate enters the cycle to make glycine and eventually pyruvate. In the CBB cycle, formate oxidation yields ATP, and the CO₂ is then captured into the CBB." The authors should write these helper sentences in the introduction as if the reader has not read every other paper on this topic, in order to make it a more valuable standalone literature reference.
5. Rubisco, would be more appropriately written as RuBisCO and requires at least one full acronym description in its first mention.
6. Line 67-68 – mention of the organism's use as agricultural protein feed must be elaborated. Is this microbe GRAS for consumption? Does it make large amounts of biomass per unit volume time? What makes it special, other than it can produce biomass from formate?
7. *C. necator* has changed names several times, some mention of this in the introduction would be important.

Scientific questions for the authors:

1. Why is a CO₂ atmosphere needed? Should not the CO₂ formed from formate oxidation result in biomass formation?
2. From a space-time yield of biomass per liter, how does formate consumption in the newly engineered strain compare to photolithoautotrophic rates in photosynthetic organisms like green algae or cyanobacteria? Would FDH expression in an alga and formate feeding result in faster heterotrophic growth, yes it requires more ATP per cycle, but what would be the benefits of *C. necator* over other CBB containing hosts? This should be discussed in the main text.

Reviewer #3 (Remarks to the Author):

In this study, the authors continued their work from the previous study (<https://doi.org/10.1016/j.ymben.2020.08.004>) to improve the strength of synthetic glycine pathway in *Cupriavidus necator*, an aerobic formatotroph. By using Tn5 transposon machinery, modules of rGly pathway were suitably integrated and expressed in *Cupriavidus necator* that enhanced the efficiency of formate use in biomass formation compared to the prior strain. The authors have conducted fundamental experiments to comprehend all strains generated from engineering process. the authors supplied proteomic data to gain a thorough understanding of the distribution of the synthetic rGly pathway. However, the authors should improve the discussion rather than only describe the results. Moreover, below are some comments to help the manuscript to be understood better.

Comments:

- 1) Please provide the information on various promoters used in this study: P14, PphaC, P3, P4, P2.
- 2) Please describe how the recombinant plasmids pC1 and pC3 were cured from the engineered cells.
- 3) Please describe briefly the CRG4 strain at the beginning of the Result section to be understood better.
- 4) Formatotrophic growth in this study needs both formate and carbon dioxide (or bicarbonate) as carbon sources. Thus,

calculation of the biomass yield based on gram cell dry weight per mole formate might be not adequate. What about the difference between consumption and generation of carbon dioxide (or bicarbonate) by the cell ?

5) Is there any evidence for the inhibition of molybdenum-dependent formate dehydrogenase by addition of tungsten ?

6) It would be great if the authors compare the efficiency of synthetic reductive glycine pathway assembled in *Cupriavidus necator* and in other bacterial hosts from the literatures.

7) The authors claimed the more efficient formate-to-biomass process via implementation of the synthetic reductive glycine pathway; however, the doubling time of the rGly-utilizing cell was lower than the CBB-utilizing cell. Is there any considerable strategy to improve the doubling time of rGly-utilizing cells?

8) As polyhydroxybutyrate (PHB) is one of the practical products of *Cupriavidus necator*, efficiently production of PHB from C1 sources like formate and CO₂ using this host strain would be a wonderful strategy, which would bring economic benefit. However, the integration of module 1 of the rGly pathway at the phasin position might negatively affect the production of polyhydroxybutyrate, what should you deal with this situation?

** I suggest that you consider Nature Communications as a suitable venue for your work. To transfer your manuscript there, please use our manuscript transfer portal. You will not have to re-supply manuscript metadata and files, unless you wish to make modifications, but please note that this link can only be used once and remains active until used. For more information, please see our [manuscript transfer FAQ](http://www.nature.com/authors/author_resources/transfer_manuscripts.html?WT.mc_id=EMI_NPG_1511_AUTHORTRANSF&WT.ec_id=AUTHOR) page.

Note that any decision to opt in to In Review at the original journal is not sent to the receiving journal on transfer. You can opt in to [In Review](https://www.nature.com/nature-portfolio/for-authors/in-review) at receiving journals that support this service by choosing to modify your manuscript on transfer. In Review is available for primary research manuscript types only.

Version 1:

Reviewer comments:

Reviewer #1

(Remarks to the Author)

All the major comments are appropriately addressed, and the revised manuscript is significantly improved. However, there are some points that need to be addressed.

1. Line 511: The reference cited for the values used for growth-associated maintenance is incorrect. Please check the reference numbering for the entire article.
2. When comparing methylotrophic yields and growth rates in Supplementary Table 7, it seems that a recent publication reporting synthetic methylotrophic *E. coli* with a doubling time of 3.5 hours is missing (<https://doi.org/10.1038/s41467-024-53206-4>).
3. Could the authors explain why they measured biomass yield in a chemostat by controlling the growth rate at a doubling time of 14 hours? Based on Fig. 2, both the CBB strains (T_d = 6) and rGlyP strains (T_d = 11) can grow faster than this controlled growth rate.

Reviewer #2

(Remarks to the Author)

The authors have adequately addressed all my comments - the article is fit for acceptance

Reviewer #3

(Remarks to the Author)

Generally, an increase in carbon dioxide assimilation efficiency leads to an accelerated cell growth rate. However, in this study, the doubling time was found to increase instead, which emerged as a critical technical issue during the review process. To address this, chemostat experiments were conducted under conditions of same growth rates, demonstrating a 17% increase in carbon dioxide assimilation efficiency. This result effectively addresses the raised concern and provides a valid response. Other comments to the previous manuscript is properly revised in the revised manuscript. Thus, I

recommend this manuscript to be published without further revision.

Decision Letter:

Our ref: NMICROBIOL-22123064A-Z

11th December 2024

Dear Nico,

Thank you for submitting your revised manuscript "Engineered synthetic one-carbon fixation exceeds yield of the Calvin Cycle" (NMICROBIOL-22123064A-Z). I apologize that this has taken a bit longer than normal, but despite the delay I'm pleased to be able to get back to you with good news. Your paper has now been seen by the original referees and their comments are below. The reviewers find that the paper has improved in revision, and therefore we'll be happy in principle to publish it in Nature Microbiology, pending minor revisions to satisfy the referees' final requests and to comply with our editorial and formatting guidelines.

Thank you again for your interest in Nature Microbiology Please do not hesitate to contact me if you have any questions.

Sincerely,

Reviewer #1 (Remarks to the Author):

All the major comments are appropriately addressed, and the revised manuscript is significantly improved. However, there are some points that need to be addressed.

1. Line 511: The reference cited for the values used for growth-associated maintenance is incorrect. Please check the reference numbering for the entire article.
2. When comparing methylotrophic yields and growth rates in Supplementary Table 7, it seems that a recent publication reporting synthetic methylotrophic *E. coli* with a doubling time of 3.5 hours is missing (<https://doi.org/10.1038/s41467-024-53206-4>).
3. Could the authors explain why they measured biomass yield in a chemostat by controlling the growth rate at a doubling time of 14 hours? Based on Fig. 2, both the CBB strains ($T_d = 6$) and rGlyP strains ($T_d = 11$) can grow faster than this controlled growth rate.

Reviewer #2 (Remarks to the Author):

The authors have adequately addressed all my comments - the article is fit for acceptance

Reviewer #3 (Remarks to the Author):

Generally, an increase in carbon dioxide assimilation efficiency leads to an accelerated cell growth rate. However, in this study, the doubling time was found to increase instead, which emerged as a critical technical issue during the review process. To address this, chemostat experiments were conducted under conditions of same growth rates, demonstrating a 17% increase in carbon dioxide assimilation efficiency. This result effectively addresses the raised concern and provides a valid response. Other comments to the previous manuscript is properly revised in the revised manuscript. Thus, I recommend this manuscript to be published without further revision.

Point-by-point rebuttal

Reviewer #1 (Remarks to the Author):

Dronsella et al. demonstrated an engineered strain of *Cupriavidus necator*, utilizing a synthetic, more energy-efficient reductive glycine pathway (rGlyP) rather than the native Calvin-Benson-Bassham (CBB) cycle, resulting in higher biomass yield than the wild type while assimilating formic acid. Building on their previous study (ref 21), which used a plasmid-based approach to introduce the rGlyP pathway, the present manuscript employed a Tn5-based transposon to avoid the plasmid burden. While this result is encouraging, this reviewer has some concerns regarding the present manuscript, which are detailed below.

We thank the reviewer for their efforts and are happy the reviewer acknowledges the encouraging results. We have now addressed the concerns raised by the reviewer by additional experiments and revisions as detailed below.

This work uses the same rGlyP as the previous publication (ref. 21), but the previous attempt did not generate a better biomass yield than the WT strain. The authors explained this difference by plasmid burden. However, there is no convincing data to support his argument.

We agree with the reviewer's comment that plasmid burden is not the proper explanation. Reductions in expression burden (e.g. plasmid burden) can lead to increases in growth rate, as we show for the genome-integrated strains created in this manuscript. These strains with reduced plasmid/expression burden grow faster than the plasmid-based strains presented in our previous work. Relatedly, they can also show a higher yield at these higher growth rates, as increased growth rates typically lead to less relative impact of energy maintenance and hence higher yields.

However, the reviewer is right that the theoretical yield improvement of the rGlyP over the CBB cycle is of stoichiometric nature and should not be related to the growth rate and hence expression burden change. Therefore, prompted by the reviewer's comments we determined the biomass yields on formate for both the final genome-integrated rGlyP and CBB cycle strain at the same controlled growth rate in a bioreactor.

These new experiments demonstrated that the yield at the same growth rate (~14 hours doubling time) of the rGlyP (4.52 gCDW/mol formate) is 17% higher than for the CBB cycle (3.88 gCDW/mol formate). These newly measured yields are higher than the ones we previously reported in batch flasks and now also closely match the absolute theoretical yields as predicted

by the genome-scale stoichiometric model (4.54 gCDW/mol formate and 3.92 gCDW/mol for rGlyP and CBB cycle respectively). With this new evidence, we now fully prove the stoichiometric advantage of the synthetic pathway in a living cell.

We extensively discuss this new evidence and alternative hypotheses in the revised manuscript (in the introduction and results sections, lines 127-145, 225-229, 287-317.)

To our understanding, the limiting enzyme in the CBB pathway, Rubisco, is a slow CO₂ fixing enzyme. rGlyP strains theoretically should grow faster than CBB strains, but the result demonstrates the opposite. Please explain.

We thank the reviewer for raising this interesting point. The rGlyP has been designed to be more stoichiometrically efficient in terms of energy (ATP) consumption than the CBB cycle, which should result in improved biomass yield. However, the growth rate, assuming the assimilation pathway is rate-limiting, is based on the pathway kinetics. RuBisCO is indeed a relatively slow enzyme with *C. necator*'s form I Rubisco catalyzing around 2-3 carboxylations per second. However, the overall pathway kinetics and growth rates are determined by the kinetics of all enzymes in the pathway. Unfortunately, many kinetic parameters for enzymes in the CBB, and especially the rGlyP are unknown. This makes it hard to predict if the rGlyP can also mediate a higher growth rate than the CBB. Based on the limitedly available data Löwe et al. (2021) predicted that the overall pathway kinetics of the rGlyP for growth on formate are likely similar to the CBB cycle¹.

In the additional experiments performed for this revision in controlled bioreactors chemostats we could demonstrate a faster growth rate than we showed previously at a doubling time of ~8 hours. Another recent study on the rGlyP in *E. coli* found a doubling on formate of 6 hours (but still a lower yield of 3.3 gCDW/mol)². These rates are still slower than the fastest growth rates achieved for the CBB on formate (~3-4 hours). However, they are in similar range. This overall suggest that the rGlyP may, after long-term adaptive evolution, support similar growth rates, to the CBB cycle. However, the key advantage of the pathways is the yield, as for the first time demonstrated in this manuscript. Yield will be the key determinant for economic feasibility and sustainability of electricity-based microbial bioproduction.

We now included a short discussion on this matter in the manuscript: "lines 333-335"

Although biomass yield and growth rate are two different parameters, they roughly correlate with each other in a non-linear fashion. However, the rGlyP shows opposite effects on the biomass and the growth rate. Please explain.

We agree with the reviewer that increased growth rates typically lead to higher yields, as we also explained above. From our data the yield of the rGlyP strains (CRG5 and CRG6) also increase, as does their growth rate. So, we are not entirely sure what opposite effect the reviewer exactly refers to. In general, we have revised the manuscript to better explain growth rate and yield effects and the main yield comparison is now done at the same growth rate in bioreactors as explained above.

The authors reported that their engineered strain reached a 14% higher biomass yield than the CBB cycle-utilizing wild type. However, Calvey et al. (2023) reported that the wild type exhibited 1.15x to 2.18x faster maximum growth rates and 10%–34% greater maximum optical density at OD600 after serial passages. It should be noted that the engineered strain and wild-type control in this manuscript have not been compared on the same basis, as the engineered strain has been passaged while the wild-type strain has not.

The paper by Calvey *et al.*, which we now refer to is an interesting study that performed long-term evolution on the *C. necator* Calvin cycle strains for faster growth on formate. They indeed achieved faster growth rates and observed higher optical densities. However, they also performed dry weight analysis; this analysis showed that the biomass yield (CDW) of their evolved strains actually did not improve. Hence, we are confident that the biomass yield improvement observed in our engineered strains is a consequence of the stoichiometric superiority of the rGlyP over the CBB cycle, and that our wildtype strain is a valid reference for that comparison.

Can the authors prove that the biomass yield increase is a consequence of rGlyP instead of serial passages? Perhaps GRC4 strain after a few passages can also reach higher biomass yield. Points No.4 &5 cast some doubts about the superiority of the rGlyP pathway. Please explain.

In line with the comments above, passaging (laboratory evolution) can indeed improve the growth rate of strains and hence the yield. However, in this revised work we measured the yield at the same controlled growth rates. So potential mutations that may increase the growth rates of the rGlyP strains do not explain the yield difference with the non-passaged CBB cycle strain grown at the same growth rate.

The authors claimed theoretical maximum biomass yield are derived from constraint-based metabolic modelling. Can you elaborate more about GAM and NGAM, and how to get these values?

We realize that this point was not clear in the original manuscript, so we have clarified the corresponding explanation in materials and methods.

Lines (509-521):

"The growth associated maintenance (GAM, 135 mmol gCDW⁻¹) and non-growth associated maintenance (NGAM, 3.0 mmol gCDW⁻¹ h⁻¹) values used to run the simulations were retrieved from Jahn et al⁴⁸. More information on their fine-tuned GAM parameter can be found between lines 101 and 104 of the script called 'run_simulations.py' within the reported Gitlab repository (https://github.com/m-jahn/genome-scale-models/blob/master/Ralstonia_eutropha/).

To calculate the theoretical biomass yield, we constrained the biomass reaction to the growth rate corresponding to 14 hours doubling time, the growth rate used in the bioreactors (Fig 4). The formate uptake rate reaction ("EX_formate_e") was used as the objective function and fluxes were computed for each scenario under the specified conditions. The ratio between growth rate (h⁻¹) and the maximum predicted formate uptake rate (mmol gCDW⁻¹ h⁻¹) was used to calculate the biomass yields in gCDW mol⁻¹ formate. Finally, we calculated the relative gain in biomass yield (%) obtained with the rGlyP compared to the CBB cycle at those conditions."

Reviewer #2 (Remarks to the Author):

Dronsella et al, Engineered synthetic one-carbon fixation exceeds yield of the Calvin Cycle – Review Nature Microbiology

In this manuscript the authors use transposable element genomic integration of 6 enzymes which were previously harbored on plasmids to show increased efficiency of the rGly pathway to formate fixation to biomass in *Cupriavidus necator*. Screening after transposable element integration and plasmid curing was performed by analyzing growth rates to identify fast growing strains fed by formate and in a CO₂ enriched environment. The integrative elements cause knockout of redundant genes and also allow high transcription rates of the target genes that, through proteomics, matched amino acid amounts of those achieved from plasmid expression of the target pathway. The authors show, for the first time, that this engineered pathway to carbon fixation can outcompete the native fixation of CO₂ to biomass from the ATP derived from formate oxidation (to CO₂) in this organism by its inherent CBB cycle.

The work is of high relevance to the microbial engineering community and should of course be published. There are some small changes which would make the manuscript more accessible to the reader which I would propose be made prior to acceptance.

We are happy the reviewer agrees on the high relevance of this work and supports publication and are thankful for the suggested changes.

CBB is abbreviated in its first appearance in the introduction, I believe the abstract does not count as a first appearance for the full words and should be repeated here the first time.

We have amended this issue and have now changed the sentence in line 23 to introduce the CBB abbreviation properly.

A figure illustrating formate use through the CBB cycle should be included with the synthetic carbon fixation cycle so the reader can instantly see the differences and intuitively understand the advantage of the rGly pathway.

We thank the reviewer for the suggestion and have now included a Calvin cycle schematic with highlighted differences in ATP consumptions, enzymatic steps, thermodynamic driving force and predicted biomass yield as compared to the reductive glycine pathway. See figure 1a,b. We have also included schematics for the generation of ATP from NADH via oxidative phosphorylation and generation of NADPH using the membrane-bound transhydrogenase reaction.

The introduction mentions anaerobic and aerobic rGly, but does not specify which the engineered one here represents and why it matters.

The reductive glycine pathway in this study represents the aerobic variant as it does not employ glycine reductase, which is a highly oxygen sensitive enzyme. The anaerobic variant is even slightly more ATP-efficient (~1 ATP per pyruvate vs. 2 ATP/pyruvate) and was recently discovered to operate in some bacteria. However, for biotechnological production from formate of biomass, protein and other compounds that require a substantial amount of ATP in the biosynthesis the aerobic pathway variant is required for efficient production. The anaerobic variant of the rGlyP can only work with other electron acceptors than oxygen, such as sulphate, which is a not a widely available electron acceptor suitable for biotechnology. We now explain this difference in a bit more detail in the revised manuscript in line 102-107: "However, for the generation of energetically costly products like biomass aerobic conditions are preferred, as more ATP is available in these conditions. The anaerobic variant of the rGlyP cannot operate in aerobic conditions as the glycine reductase enzyme is not oxygen-tolerant."

The introduction could benefit from being somewhat more didactic in topic presentation: Something along the lines of “Formate oxidation generates ATP, non-oxidized formate enters the cycle to make glycine and eventually pyruvate. In the CBB cycle, formate oxidation yields ATP, and the CO₂ is then captured into the CBB.” The authors should write these helper sentences in the introduction as if the reader has not read every other paper on this topic, in order to make it a more valuable standalone literature reference.

We thank the reviewer for this suggestion. We tried to balance word count and explanation to match the journal article format. However, we agree that a few sentences to explain the overall pathway operation is helpful and have now added the following sentences in lines 83-91: “Growth on formate via the CBB cycle requires complete oxidation of formate to CO₂ and NADH. CO₂ is then re-fixed in the CBB cycle by the generated NADH and ATP (obtained from NADH via oxidative phosphorylation). In contrary, growth via the rGlyP on formate occurs via direct assimilation of formate. This assimilation, however, also still requires additional NADH and ATP, which is generated via oxidation of a large part of the formate in parallel to formate assimilation (Fig. 1a). This formation oxidation reaction in addition also supplies a stoichiometric surplus of CO₂. This CO₂ generation can support elevated CO₂ levels in the cell to drive the thermodynamically reversible carboxylation reaction of the glycine cleavage system (GCV) in assimilation pathway.”

Rubisco, would be more appropriately written as RuBisCO and requires at least one full acronym description in its first mention.

We thank the reviewer for this oversight and have now properly introduced RuBisCO as ribulose-1,5-bisphosphate carboxylase/oxygenase in line 60 and henceforth refer to it as RuBisCO in the text.

Line 67-68 – mention of the organism’s use as agricultural protein feed must be elaborated. Is this microbe GRAS for consumption?

We thank the reviewer for raising this point. While *Cupriavidus necator* is not formally labelled by the US FDA as Generally Regarded As Safe, it received the QPS (Qualified Presumption of Safety) status by the European Food and Safety Authority (EFSA) and is considered safe to use in food as dead cells (whole biomass). We included a reference in line 123 to an article by EFSA discussing this. Also, several companies are investigating and developing *C. necator* and related bacteria for food and feed (e.g. Deep Branch, Solar Foods and Circe).

Does it make large amounts of biomass per unit volume time? What makes it special, other than it can produce biomass from formate?

C. necator is a one of the fastest growing organisms on formate. Together with the yield improvement demonstrated in this article, this can indeed lead to high volumetric productivities (space time yield) of biomass. In addition, *C. necator* is developing into a platform organism with an increasing genetic toolbox, which has been already engineered for a range of products³⁻⁶. and has been previously demonstrated up to industrial scale for native production of polyhydroxybutyrate (PHB)⁷. Altogether, this makes it the most attractive platform organism known for formatotrophic production. By demonstrating the more efficient synthetic formate fixation pathway in this work, the future potential of this platform species has been further expanded.

C. necator has changed names several times, some mention of this in the introduction would be important.

This is indeed a good point raised by the reviewer, we now mention the best known two previous names (*Ralstonia eutropha* and *Alcaligenes eutrophus*) for this organisms in lines 116-118: "*Cupriavidus necator* (formerly known also by different names including *Ralstonia eutropha* and *Alcaligenes eutrophus*) is an aerobic bacterium that natively utilizes the CBB cycle for formatotrophic growth."

Scientific questions for the authors:

Why is a CO₂ atmosphere needed? Should not the CO₂ formed from formate oxidation result in biomass formation?

The reviewer is correct that formate oxidation, which is essential for growth on the rGlyP, leads to intracellular CO₂ generation. This could indeed lead to higher internal CO₂ concentrations, which could support the glycine synthesis reaction in the rGlyP, which requires elevated CO₂. However, in the open headspace batch flasks and highly aerated bioreactors cultures done in this manuscript, which both contained relatively low cell density, internally generated CO₂ quickly escapes and likely cannot support this activity. However, for the upscaling of rGlyP -based growth of *C. necator* in high cell density cultures the internal CO₂ may indeed be sufficient to support the rGlyP. This is an interesting remark, which we briefly now refer to in the manuscript line 388-392" Strictly seen enough CO₂ is generated intracellularly for driving the GCV carboxylation of the rGlyP by formate oxidation. However, in relatively low biomass density cultures, with high

aeration, as performed in this study, this may be insufficient. Hence, we supplement bicarbonate in the medium and 10 % CO₂ in the headspace/ gas supply during the bioreactor cultivation.”

From a space-time yield of biomass per liter, how does formate consumption in the newly engineered strain compare to photolithoautotrophic rates in photosynthetic organisms like green algae or cyanobacteria? Would FDH expression in an alga and formate feeding result in faster heterotrophic growth, yes it requires more ATP per cycle, but what would be the benefits of *C. necator* over other CBB containing hosts? This should be discussed in the main text.

The maximum growth rates of some fast-growing cyanobacteria and microalgae can be higher than the formatotrophic growth rates of *C. necator*. However, photoautotrophic space-time yields are largely limited by light-penetration and self-shading and hence can only efficiently occur in reactors with large surface areas and limited depth (Lips, 2018⁸). Formatotrophic growth can be performed in more common reactor tanks with a much higher space-time yield per m² reactor footprint. This is an interesting remark, and we briefly refer to this in the text in line 39-42:“ An alternative is provided by microbial bioproduction using photosynthetic microorganisms such as cyanobacteria and microalgae, which can also be grown on non-arable land. However, these microbial photosynthetic approaches do not scale well due to issues associated light penetration and self-shading.“

To increase the volumetric productivities of photoautotrophs they could indeed be grown mixotrophically using both light and formate as energy source, when they are engineered with a FDHs. We have proposed this before in an opinion article, and proof-of-principle FDH expression and improved growth with formate has been recently shown in microalgae (Dahlin, 2023)⁹. However, in that case the formatotrophic/phototrophic growth would still rely on the native Calvin cycle present in all microalgae and cyanobacteria, and still photobioreactors would be needed that require a high surface area. Hence, we believe this is not as effective as realizing efficient formatotrophic growth via the rGlyP and we consider this alternative out of the scope for this article.

Reviewer #3 (Remarks to the Author):

In this study, the authors continued their work from the previous study (<https://doi.org/10.1016/j.ymben.2020.08.004>) to improve the strength of synthetic glycine pathway in *Cupriavidus necator*, an aerobic formatotroph. By using Tn5 transposon machinery, modules of rGly pathway were suitably integrated and expressed in *Cupriavidus necator* that enhanced the efficiency of formate use in biomass formation compared to the prior strain. The

authors have conducted fundamental experiments to comprehend all strains generated from engineering process. the authors supplied proteomic data to gain a thorough understanding of the distribution of the synthetic rGly pathway. However, the authors should improve the discussion rather than only describe the results. Moreover, below are some comments to help the manuscript to be understood better.

We thank the reviewer for their conclusion that we provided thorough results. We are also thankful for the provided suggestions that improved our discussion section.

Comments:

Please provide the information on various promoters used in this study: P14, P_{PhaC}, P3, P4, P2.

We briefly describe these different strengths constitutive promoter (also used in our previous work) now better in the text in lines 418-419 as follows: " The operons *mtdA-fch-fftL* and *sdaA-glyA* were previously cloned under control of the promoters, from weakest to strongest, P₁₄/P_{PhaC}/P₃/P₄/P₂ and P_{cat}/P_{PhaC}/P₃/P₄ respectively. "

Please describe how the recombinant plasmids pC1 and pC3 were cured from the engineered cells.

This we indeed forgot to include, and we now included steps for plasmid curing the methods section in line 405-411 as follows: "Cells containing plasmids to be cured were propagated in LB media without antibiotics. From each grown passage 50 colonies were streaked out on LB agar plates. These were then replica plated on LB agar plates containing the antibiotic for which the plasmid would provide resistance and on non-antibiotic plates. Once colonies were obtained that did not grow on the antibiotic containing plates, the respective colonies from the non-selective plate were investigated via PCR targeting the plasmid to confirm the curing of the clone."

Please describe briefly the CRG4 strain at the end of the introduction section to be understood better.

Thanks for this suggestion. We briefly introduce the strain now at the beginning of the results section in line 127-129 as follows: "Our previous study implemented the aerobic variant of the rGlyP (CRG4) in *C. necator*, by expressing modules C1 and C3 from plasmid, the C2 module from the genome and performing ALE until a doubling time of 12 h was reached. "

Formatotrophic growth in this study needs both formate and carbon dioxide (or bicarbonate) as carbon sources. Thus, calculation of the biomass yield based on gram cell dry weight per mol

formate might be not adequate. What about the difference between consumption and generation of carbon dioxide (or bicarbonate) by the cell?

The reviewer is correct that the rGlyP consumes both formate and CO₂, however, all energy (electrons) are supplied from the formate. In addition, the small amount of CO₂ required in the pathway can be supplied by formate as a major fraction of formate is oxidized to supply electrons and this process also co-generates an excess of CO₂. In the current small scale, low cell density set-up this CO₂ can quickly escape, keeping intracellular CO₂ levels too low and requiring co-sparging of CO₂ (see also answer to 'Scientific question to the authors #1' raised by reviewer 2). Overall, as CO₂ is not an energy source and there is no net CO₂ consumption in the biological process, hence the yield per formate is the right metric to evaluate the pathway.

Is there any evidence for the inhibition of molybdenum-dependent formate dehydrogenase by addition of tungsten?

Yes, this has been shown in literature, specifically in the study of Friedbold and Bowien in 1997¹⁰, which we cited in our study. In the revised version we further clarified this in line 269 and 270 as follows: "To test this, we gradually inhibited molybdenum-dependent sFDH activity via competitive titration with the non-active metal tungsten, based on previously demonstrated methods."

It would be great if the authors compare the efficiency of synthetic reductive glycine pathway assembled in *Cupriavidus necator* and in other bacterial hosts from the literatures.

We like this suggestion of the reviewer and added Supplementary Table 7 providing an overview of all available growth yield and rates on formate for the different organisms in which the CBB, rGlyP, RuMP and XuMP has been implemented so far. We refer to this table in lines 337-340: "In fact, the measured formatotrophic yield of 4.52 gCDW/mol is to our best knowledge higher than any so far reported yield on formate for natural organisms using the CBB cycle, as well as for engineered formatotrophs (Supplementary Table 7)."

The authors claimed the more efficient formate-to-biomass process via implementation of the synthetic reductive glycine pathway; however, the doubling time of the rGly-utilizing cell was lower than the CBB-utilizing cell. Is there any considerable strategy to improve the doubling time of rGly-utilizing cells?

This is a good point raised by the reviewer. In the bioreactor experiments on our most advanced strains (CRG6 Δ *dadA6*) we could now also demonstrate the faster rGlyP growth rate for *C. necator* down to ~8 hours doubling time. However, the growth rate could potentially be

improved further and potentially also faster than the Calvin cycle (~3-4 hours doubling time). However, if apart from the yield also the rate can be beaten by the rGlyP remains an open question. This will depend on the maximum overall kinetics of all enzymes in the rGlyP versus the overall kinetics of the Calvin Cycle. Based on the best possible predictions so far, these rates will be similar (Lowe et al. 2021)¹. However, to further improve the growth rate long-term evolution experiments should be performed, we now briefly discuss this option in the discussion in lines 333-335 as follows: “The genome integrated rGlyP strain generated in this study, allowed for improved growth of *C. necator* via this synthetic pathway down to ~8 hours doubling time. This growth rate may be further improved by long-term evolution as shown recently for the wild-type strain.”

As polyhydroxybutyrate (PHB) is one of the practical products of *Cupriavidus necator*, efficiently production of PHB from C1 sources like formate and CO₂ using this host strain would be a wonderful strategy, which would bring economic benefit. However, the integration of module 1 of the rGly pathway at the phasin position might negatively affect the production of polyhydroxybutyrate, what should you deal with this situation?

We agree with the reviewer that PHB is interesting for bioproduction. For efficient production of PHB, the interrupted gene encoding the PhaP1 protein is indeed relevant as it has been shown to increase final PHB yields¹¹. Further engineering of rGlyP strains for optimal PHB production would hence require the insertion of this gene back into the genome or via a plasmid, as could also be done for the introduction of other (non-native) production routes.

References:

1. Löwe, H. & Kremling, A. In-Depth Computational Analysis of Natural and Artificial Carbon Fixation Pathways. *BioDesign Res.* **2021**, 9898316 (2021).
2. Kim, S. *et al.* Optimizing *E. coli* as a formatotrophic platform for bioproduction via the reductive glycine pathway. *Front. Bioeng. Biotechnol.* **11**, 1091899 (2023).
3. Li, H. *et al.* Integrated Electromicrobial Conversion of CO₂ to Higher Alcohols. *Science* **335**, 1596–1596 (2012).
4. Milker, S. *et al.* Gram-scale production of the sesquiterpene α -humulene with *Cupriavidus necator*. *Biotechnol. Bioeng.* **118**, 2694–2702 (2021).

5. Collas, F. *et al.* Engineering the biological conversion of formate into crotonate in *Cupriavidus necator*. *Metab. Eng.* **79**, 49–65 (2023).
6. Müller, J. *et al.* Engineering of *Ralstonia eutropha* H16 for Autotrophic and Heterotrophic Production of Methyl Ketones. *Appl. Environ. Microbiol.* **79**, 4433–4439 (2013).
7. Carpine, R., Olivieri, G., Hellingwerf, K. J., Pollio, A. & Marzocchella, A. Industrial Production of Poly- β -hydroxybutyrate from CO₂: Can Cyanobacteria Meet this Challenge? *Processes* **8**, 323 (2020).
8. Lips, D., Schuurmans, J. M., Branco Dos Santos, F. & Hellingwerf, K. J. Many ways towards ‘solar fuel’: quantitative analysis of the most promising strategies and the main challenges during scale-up. *Energy Environ. Sci.* **11**, 10–22 (2018).
9. Dahlin, L. R. *et al.* Heterologous expression of formate dehydrogenase enables photoformatotrophy in the emerging model microalga, *Picochlorum renovo*. *Front. Bioeng. Biotechnol.* **11**, (2023).
10. Friedebold, J. & Bowien, B. Physiological and biochemical characterization of the soluble formate dehydrogenase, a molybdoenzyme from *Alcaligenes eutrophus*. *J. Bacteriol.* **175**, 4719–4728 (1993).
11. Tang, R., Peng, X., Weng, C. & Han, Y. The Overexpression of Phasin and Regulator Genes Promoting the Synthesis of Polyhydroxybutyrate in *Cupriavidus necator* H16 under Nonstress Conditions. *Appl. Environ. Microbiol.* **88**, e01458-21 (2022).

Point-by-point rebuttal

Reviewer #1:

Remarks to the Author:

All the major comments are appropriately addressed, and the revised manuscript is significantly improved. However, there are some points that need to be addressed.

We thank the reviewer for their support.

1. Line 511: The reference cited for the values used for growth-associated maintenance is incorrect. Please check the reference numbering for the entire article.

We have indeed misplaced this reference and have now checked and updated all references.

2. When comparing methylotrophic yields and growth rates in Supplementary Table 7, it seems that a recent publication reporting synthetic methylotrophic *E. coli* with a doubling time of 3.5 hours is missing (<https://doi.org/10.1038/s41467-024-53206-4>).

The reviewer is right in that we only had referenced that BioRxiv article but not yet the peer-reviewed publication, which was published recently. We now reference the latter one.

3. Could the authors explain why they measured biomass yield in a chemostat by controlling the growth rate at a doubling time of 14 hours? Based on Fig. 2, both the CBB strains ($T_d = 6$) and rGlyP strains ($T_d = 11$) can grow faster than this controlled growth rate.

As the reviewer pointed out we could have measured the biomass yields at a slightly faster dilution rate. However, we decided to go for a dilution rate of 0.05 h^{-1} (equal to 14 h doubling time) to make our results more comparable to the most relevant publication on the biomass yield of *C. necator* on formate in chemostat from Grunwald et al., 2015. This allowed us to assess how close our CBB cycle employing wildtype-like PHB- *C. necator* (ΔphaC1) strain came to the performance of the a similar PHB- *C. necator* strain used in the paper Grunwald et al.. Here we found a yield reached $3.88 \text{ gCDW/mol formate}$ compared to their 4.056 gCDW/mol (0.16 mol/mol) formate. With this in hand we were all the more confident that our yield of $4.52 \text{ gCDW/mol formate}$ for the synthetic reductive glycine pathway strain was not only higher relative to “our” wildtype like strain, but also well above this established literature reference point. We also explained the logic for this dilution rate accordingly in the Main text.

Reviewer #2:

Remarks to the Author:

The authors have adequately addressed all my comments - the article is fit for acceptance

We thank the reviewer for their support.

Reviewer #3:

Remarks to the Author:

Generally, an increase in carbon dioxide assimilation efficiency leads to an accelerated cell growth rate. However, in this study, the doubling time was found to increase instead, which emerged as a critical technical issue during the review process. To address this, chemostat experiments were conducted under conditions of same growth rates, demonstrating a 17% increase in carbon dioxide assimilation efficiency. This result effectively addresses the raised concern and provides a valid response. Other comments to the previous manuscript is properly revised in the revised manuscript. Thus, I recommend this manuscript to be published without further revision.

We thank the reviewer for his helpful comments and the support.